# Leukemia inhibitory factor suppresses hepatic de novo lipogenesis and induces cachexia in mice

Xue Yang[1], Jianming Wang [1], Chun-Yuan Chang [1], Fan Zhou[1], Juan Liu[1], Huiting Xu[2], Maria Ibrahim[3], Maria Gomez[3], Grace L. Guo[4,5,6], Hao Liu[7,8], Wei-Xing Zong[3,9], Fredric E. Wondisford[2], Xiaoyang Su [2,10], Eileen White [3,11], Zhaohui Feng [1] ✉ & Wenwei Hu [1] ✉

Cancer cachexia is a systemic metabolic syndrome characterized by involuntary weight loss, and muscle and adipose tissue wasting. Mechanisms underlying cachexia remain poorly understood. Leukemia inhibitory factor (LIF), a multi-functional cytokine, has been suggested as a cachexia-inducing factor. In a transgenic mouse model with conditional LIF expression, systemic elevation of LIF induces cachexia. LIF overexpression decreases de novo lipogenesis and disrupts lipid homeostasis in the liver. Liver-specific LIF receptor knockout attenuates LIF-induced cachexia, suggesting that LIF-induced functional changes in the liver contribute to cachexia. Mechanistically, LIF over-expression activates STAT3 to downregulate PPARα, a master regulator of lipid metabolism, leading to the downregulation of a group of PPARα target genes involved in lipogenesis and decreased lipogenesis in the liver. Activating PPARα by fenofibrate, a PPARα agonist, restores lipid homeostasis in the liver and inhibits LIF-induced cachexia. These results provide valuable insights into cachexia, which may help develop strategies to treat cancer cachexia.

Many cancer patients, especially those in advanced stages, exhibit cachexia, a systemic disorder characterized by involuntary weight loss and the wasting of muscle and adipose tissue[1–3]. Cachexia occurs in up to 80% of patients with advanced cancers and accounts for 20–30% of cancer-associated deaths[1,4,5]. Cachexia also occurs in multiple chronic non-malignant diseases, including infection, chronic obstructive pulmonary disease (COPD), chronic heart failure, end-stage renal failure (ERSF), AIDS, etc.[4]. Cachexia is driven by a combination of reduced food intake, inflammation, and metabolic changes[1,4,5]. Cachexia

exhibits many metabolic changes, including perturbed energy balance and the stimulation of catabolism in multiple organs, such as the muscle, fat, liver, and heart[6]. In addition to the typical cachexia symptoms, cancer cachexia often leads to reduced tolerance and diminished therapeutic responses to chemotherapy, which further negatively impact upon the prognosis and survival of patients[1–3]. Clinical management of cancer cachexia remains challenging because of the complexity of this metabolic disorder and the lack of effective therapies. There is an urgent need to gain a better understanding of

[1]Department of Radiation Oncology, Rutgers Cancer Institute of New Jersey, Rutgers University, New Brunswick, NJ, USA. [2]Department of Medicine, Rutgers-Robert Wood Johnson Medical School, New Brunswick, NJ, USA. [3]Rutgers Cancer Institute of New Jersey, Rutgers University, New Brunswick, NJ, USA. [4]Department of Pharmacology and Toxicology, Rutgers University, Piscataway, NJ, USA. [5]Environmental and Occupational Health Science Institute, Rutgers University, Piscataway, NJ, USA. [6]Department of Veterans Affairs New Jersey Health Care System, East Orange, NJ, USA. [7]Department of Biostatistics and Epidemiology, Rutgers School of Public Health, Piscataway, NJ, USA. [8]Biostatistics Shared Resource, Rutgers Cancer Institute of New Jersey, Rutgers University, New Brunswick, NJ, USA. [9]Department of Chemical Biology, Ernest Mario School of Pharmacy, Rutgers University, Piscataway, NJ, USA. [10]Metabolomics Core Facility, Rutgers Cancer Institute of New Jersey, New Brunswick, NJ, USA. [11]Ludwig Princeton Branch, Ludwig Institute for Cancer Research, Princeton University, Princeton, NJ, USA. ✉e-mail: fengzh@cinj.rutgers.edu; wh221@cinj.rutgers.edu

the mechanisms underlying cachexia, particularly in the context of cancer.

Cancer cachexia may be in part driven by the competition between tumor and host cells for nutrients[1]. Importantly, there is metabolic and signaling crosstalk between organs, including the brain, liver, bone, gut, muscle, and adipose tissues, which contributes to whole-body wasting and the development of the cachectic state. A complex group of tumor- and host-derived inflammatory cytokines and other factors, including tumor necrosis factor-α (TNFα), interleukin-1 (IL-1), interleukin-6 (IL-6), growth/differentiation factor 15 (GDF15), etc., function as important mediators of cachexia[7]. For example, TNFα activates the ubiquitin-proteasome system (UPS) to promote muscle protein breakdown[8]. The elevation of IL-6 during cachexia accelerates muscle and fat wasting[7]. However, the dynamic change of these cytokines during cachexia development and their precise roles in cachexia remain incompletely understood, and single cytokine-targeted approaches have so far shown limited clinical benefits.

Leukemia inhibitory factor (LIF) is a multi-functional cytokine that acts through binding to its receptor complex composed of LIF receptor (LIFR) and glycoprotein 130 (gp130) to activate downstream signaling pathways[9]. LIF is frequently overexpressed in solid tumors, correlating with poor cancer patient prognosis[9,10]. LIF has been suggested as a cachexia-inducing factor. Several human and mouse cancer cell lines, including mouse colorectal carcinoma cell line C26, secrete LIF, which is associated with cachexia development in tumor-bearing mice[9,11]. For example, C26 tumor-bearing mice exhibit progressive body weight loss and poor survival, which can be largely alleviated by LIF neutralizing antibody treatment[11,12]. In vitro, recombinant LIF induces lipolysis in cultured adipocytes[12]. Peripheral LIF administered to mice leads to cachexia-associated adipose loss and body weight loss[12]. However, the role of LIF in cancer cachexia and especially its underlying mechanism are far from clear.

In this study, we generated a transgenic LIF knock-in mouse model that can conditionally induce LIF overexpression. This model allowed us to investigate the role and mechanism of LIF in cachexia. Systemic elevation of LIF levels in mice induced cachexia syndrome with the loss of muscle and adipose tissues, negative energy balance, and impaired survival. Characterizing the metabolic changes in mice during cachexia development revealed decreased hepatic de novo lipogenesis and disrupted lipid homeostasis induced by LIF overexpression in mice. The down-regulation of PPARα, a master regulator of lipid metabolism, is an important mechanism underlying the decreased hepatic de novo lipogenesis. Activating PPARα by feeding mice with the diet containing PPARα agonist fenofibrate restored lipid homeostasis in the liver and significantly inhibited cachexia induced by LIF overexpression. These results demonstrate the systemic effect of LIF on cachexia, unveil a mechanism underlying LIF-induced cachexia, and suggest that activating PPARα to restore hepatic de novo lipogenesis could serve as a potential strategy for cachexia treatment.

## Results

### LIF plays a key role in cachexia development in mice
Previous studies have shown that mice bearing C26 tumors develop cachexia[12,13], and it has been suggested that the elevated LIF levels produced by C26 tumors contribute to cachexia in the tumor-bearing mice[11]. To confirm that LIF contributes to cachexia in this model, C26 cells with *LIF* knockout (KO) by the CRISPR/Cas9 system (C26-LIF KO) were used to establish subcutaneous (*s.c.*) syngeneic xenograft tumors in Balb/c mice. LIF was highly expressed in tumors formed by C26 cells, but undetectable in tumors formed by C26-LIF KO cells (Fig. S1A). Consistent with previous reports[11–13], Balb/c mice bearing C26 cell-formed tumors exhibited symptoms of cachexia shortly after tumor formation. Mice bearing tumors at the size of ~400 mm³ showed a significant decrease in body weight (Fig. S1B), muscle wasting, and

white adipose tissue (WAT) loss as examined by H&E staining (Fig. S1C), which are main characteristics of cachexia, and these mice had a median survival of 15 days (Fig. S1D). Notably, *LIF* knockout in C26 cells greatly mitigated cachexia in tumor-bearing mice. Compared with C26 tumors, the growth of C26-LIF KO tumors was much slower (Fig. S1E). When tumors reached a comparable size, mice bearing C26-LIF KO tumors showed less body weight loss, and reduced muscle and WAT wasting compared with mice bearing C26 tumors (Fig. S1B, C). Consequently, mice bearing C26-LIF KO tumors had a much longer survival than C26 tumor-bearing mice; the majority of mice bearing C26-LIF KO tumors reached the humane endpoint due to tumor size, while only a small percentage of these mice reached the humane endpoint due to cachexia (Fig. S1D). These results demonstrate that high LIF levels produced from C26 tumor cells play a crucial role in cachexia, validating LIF as a crucial tumor-produced cachexia factor.

LIF plays a profound role in promoting the proliferation, growth, survival, and metabolic reprogramming in many solid tumor cells[9,10,14]. Currently, it remains unclear how LIF promotes cachexia. The C26 tumor-induced cachexia model cannot differentiate the contribution of the direct effect of LIF secreted from tumor cells on cachexia from the secondary effects of LIF in promoting the growth, proliferation and nutritional needs of tumor cells on cachexia. To investigate whether systemic elevation of LIF induces cachexia in mice, we generated a transgenic LIF knock-in mouse model (*LIF-tg flox/+*; referred to as TgL hereafter) by knocking in the mouse *LIF* gene, preceded by the CAG promoter and a transcriptional STOP cassette, into the Rosa26 locus (CAG-STOP-LIF-eGFP-Rosa26TV) using the CRISPR/Cas9 system, as described previously[15,16] (Fig. 1A). TgL mice were then crossed with R26-Cre^ERT2 mice to generate TgL/R26-Cre^ERT2 (referred to as TgLC hereafter) mice (Fig. 1B). Eight-week-old TgLC mice were injected (intraperitoneal, *i.p.*) with tamoxifen (TAM) to induce LIF expression. Age- and gender-matched TgL mice injected with TAM served as controls. TAM injection significantly increased LIF expression levels in TgLC mice, with serum LIF levels at $330 \pm 87$ pg/mL as determined by ELISA assays (Fig. 1C). These levels are comparable with the serum LIF levels in some cancer patients, including those with pancreatic ductal adenocarcinoma (PDAC) or oesophageal adenocarcinoma (OAC), as reported previously[17–19]. Systemic elevation of the LIF levels quickly induced cachexia in TgLC mice. Mice started to lose weight at ~3 days after TAM injection in TgLC mice (Fig. 1D). After TAM injection, TgLC mice quickly lost fat mass, and subsequently started to lose lean mass at ~3 days after TAM injection, as measured by EchoMRI (Fig. 1E). At 10 days after TAM injection, the average loss of fat and lean mass was $1.63 \pm 0.51$ g and $5.58 \pm 1.35$ g, respectively (Fig. 1E). The wasting of the muscle and WAT was validated by H&E assays (Fig. 1F). Cachexia is often associated with anorexia[2,3]. A trend of a slight decrease in body weight and lean mass, and a significant decrease in fat mass were observed in the pair-fed TgL mice with TAM injection compared with the regular-fed TgL mice with TAM injection (Fig. S2A, B). However, the decrease in body weight, lean mass, and fat mass in the pair-fed TgL mice with TAM injection was significantly less than their decrease in TgLC mice with TAM injection (compare Fig. S2A, B vs. Fig. 1E).

The reduced body weight of TgLC mice after TAM injection represents a negative energy balance, indicating changes in energy intake and/or energy expenditure (EE). To characterize energy intake and EE in TgLC mice post TAM injection, we employed the Promethion Metabolic Cage System. Energy balance, calculated based on energy intake and EE, was positive during the dark cycle (19:00 pm to 7:00 am) and negative during the light cycle (7:00 am to 19:00 pm) in these mice. TAM injection significantly reduced energy balance in TgLC mice during the dark cycle but not during the light cycle (Fig. 1G). Mice consume most of their food during the dark cycle[20]. TAM injection significantly reduced food intake in TgLC mice, normalized with or without mouse body weight, during the dark cycle but not during the light cycle (Figs. 1H and S2C). The decrease in food intake during the

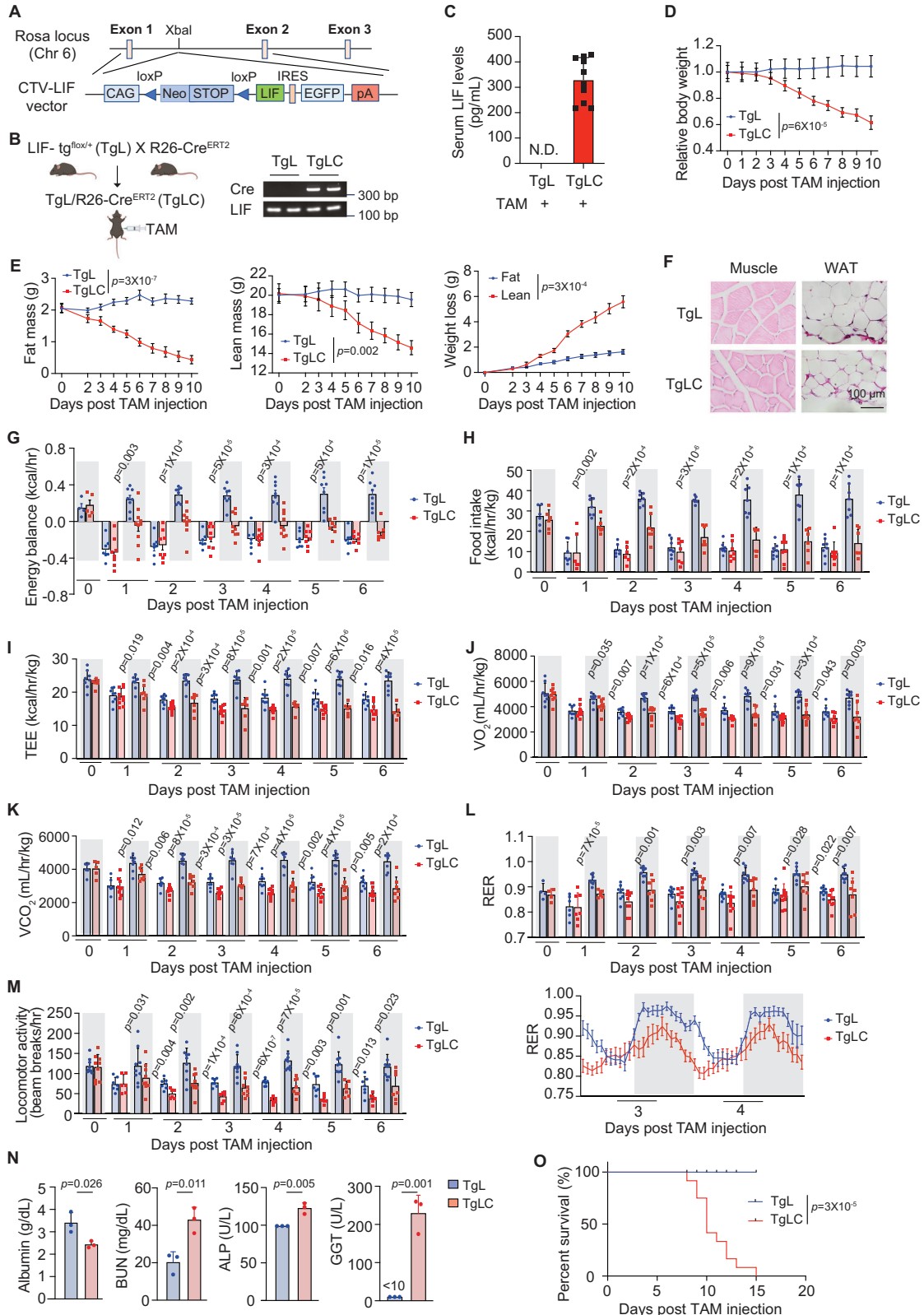

dark cycle appeared from the first dark cycle post TAM injection in TgLC mice and became more pronounced in later days (Figs. 1H and S2C).

Total EE (TEE), which reflects resting metabolism and physical activity, is calculated based on the oxygen consumption ($VO_2$) and carbon dioxide production ($VCO_2$) by an indirect calorimeter[21]. TEE was significantly reduced in TgLC mice with TAM injection, normalized

with or without mouse body weight, in both the dark and light cycles (Fig. 1I, S2D, E). Compared with TgL mice, TAM injection reduced $VO_2$ and $VCO_2$ in TgLC mice, normalized with or without mouse body weight (Figs. 1J, K and S2F–I). This reduction in both $VO_2$ and $VCO_2$ was observed in both the dark and light cycles, but it was more pronounced in the dark cycle (Figs. 1J, K and S2F–I). The respiratory exchange ratio (RER), calculated as the ratio between $VCO_2$ and $VO_2$, reflects the

**Fig. 1 | LIF overexpression induces cachexia in the TgLC mice. A** The strategy to generate *LIF-tg flox/+* (TgL) mice. The mouse *LIF* gene, preceded by the CAG promoter and a transcriptional STOP cassette, was knocked into the Rosa26 locus (CAG-STOP-LIF-eGFP-Rosa26TV) using the CRISPR/Cas9 system. **B** The generation of TgL/R26-Cre[ERT2] (TgLC) mice. Right panel: the genotyping analysis of TgL and TgLC mice by PCR. All mice with the same genotype have similar results. **C** Serum LIF levels in TgL (*n* = 6) and TgLC mice (*n* = 10) with TAM injection measured by the ELISA assay. **D** Mouse body weight post TAM injection in TgL mice (*n* = 6) and TgLC mice (*n* = 14). **E** Fat and lean mass loss post TAM injection in TgL (*n* = 10) and TgLC (*n* = 8) mice. Body composition was measured by EchoMRI. **F** Representative H&E images of muscle and WAT tissues from TgL and TgLC mice with TAM injection. At least three independent biological replicates were performed. **G–M** Mice were housed in Promethion metabolic cages. Mice were injected with TAM at the first light cycle (*n* = 4–10/group). Shaded regions represent the dark cycle from 19:00 pm to 7:00 am. Values are hourly means. Energy balance (**G**), food intake (**H**), total energy expenditure (TEE) (**I**), oxygen consumption (VO₂) (**J**), carbon dioxide production (VCO₂) (**K**), Respiratory exchange ratio (RER) (**L**) and locomotor activity (**M**) of TgL and TgLC mice post TAM injection were measured. **N** The serum levels of albumin, blood urea nitrogen (BUN), alkaline phosphatase (ALP) and gamma-glutamyl transferase (GGT) reflecting kidney and liver functions in TgL and TgLC mice at 3 days after TAM injection for albumin and 9 days after TAM injection for other parameters (*n* = 3/group). **O** Kaplan-Meier survival curves of mice. The day of TAM injection was denoted as D0. Data are presented as mean ± SEM for (**D**, **E**), and as mean ± SD for (**C**, **G–N**). N.D. non-detectable. Each dot represents an individual mouse. Both female and male mice were used. For **G–N**: Two-tailed Student's *t*-test; for **D**, **E** two-way ANOVA followed by Sidak's multiple comparison test; and for **O** two-tailed Kaplan-Meier survival analysis. Source data are provided as Source Data file.

source of metabolic fuel for EE. RER was significantly reduced in TgLC mice post TAM injection during the dark cycle (Fig. 1L), indicating a shift in metabolic fuel from carbohydrate to fat[22]. TAM injection also reduced spontaneous locomotor activity in TgLC mice during both the dark and light cycles (Fig. 1M). These data demonstrate that the reduced food intake, decreased TEE, decreased physical activity, and decreased RER in TgLC mice with LIF overexpression during cachexia development collectively led to the reduced energy balance.

Cachexia can lead to multi-organ failure[1]. LIF overexpression in TgLC mice impaired renal and liver functions, as reflected by the decreased serum levels of albumin and increased serum levels of blood urea nitrogen (BUN), which reflect renal function, and the increased serum levels of alkaline phosphatase (ALP) and gamma-glutamyl transferase (GGT), which reflect liver function, at 3 days after TAM injection for serum albumin levels and 9 days after TAM injection for other parameters (Fig. 1N). TgLC mice with TAM injection had a median survival of 10 days (Fig. 1O). Taken together, these results demonstrate that the systemic elevation of LIF levels induces cachexia in mice.

## LIF overexpression disrupts lipid homeostasis and decreases hepatic de novo lipogenesis in mice

To assess the systemic metabolic changes in TgLC mice with LIF overexpression, we performed metabolomics analyses to measure polar and lipid metabolites in the serum from TgLC and TgL mice with TAM injection. Considering the anorexia induced by LIF overexpression in TgLC mice, we included mice under both fed and fasted conditions for metabolomics analysis. A Heatmap analysis showed changes in polar metabolite levels in the serum between TgLC and TgL mice with TAM injection under both fed and fasted conditions (Fig. S3A). Under the fed condition, there was a trend indicating decreases in several metabolites involved in glucose and amino acid metabolism, including glucose, pyruvate, lactate, glycine, and glutamate in TgLC mice compared with TgL mice (Fig. S3B). A similar change was observed for some metabolites, including glucose, pyruvate, and lactate, between TgLC mice and TgL mice under the fasted condition (Fig. S3B). It has now been recognized that lipid metabolism, including triglyceride (TG) hydrolysis, is a major metabolic pathway involved in the initiation and/or progression of cancer cachexia[4,7,23]. Significant changes in lipid metabolites, notably TGs, were observed in TgLC mice with TAM injection compared with TgL mice with TAM injection under both fed and fasted conditions (Figs. 2A and S3C). These TGs are long-chain triglycerides (LC-TGs), composed of fatty acyl chains containing more than 12 carbon atoms. In TgL mice with TAM injection, under the fed condition, the majority of TGs detected in the serum had a carbon chain equal to or less than 54 (C ≤ 54) (referred to as small LC-TGs hereafter) (Fig. 2A). Fasting of TgL mice with TAM injection led to a significant decrease in TG levels in the serum (Fig. 2A, B). Notably, under the fed condition, the levels of small LC-TGs were significantly lower in TgLC mice with TAM injection compared with TgL mice with TAM injection, and the levels of LC-TGs with a carbon chain larger than 54 (C > 54) (referred to as larger LC-TGs hereafter) were significantly higher in TgLC mice with TAM injection compared with TgL mice with TAM injection (Fig. 2A, C). The changes in these TGs in TgLC mice with TAM injection were not due to food intake, as fasting in TgLC mice did not cause a significant change in the majority of TGs (Fig. 2A, B). Further identification of fatty acids (FAs) that form TGs showed that the majority of small LC-TGs were composed of saturated fatty acids (SFAs), including C14:0, C15:0, C16:0 and C18:0, and the majority of larger LC-TGs were composed of long-chain poly-unsaturated fatty acids (LC-PUFAs), including C20:4, C20:5, C22:4, C22:5, C22:6, C24:6 (Supplementary Table 1). Notably, these LC-PUFAs are essential FAs that cannot be synthesized de novo by mice, suggesting that these LC-PUFAs were mainly produced from the lipolysis of other organs[24,25]. The liver is an important organ involved in lipid homeostasis. Considering that TAM-induced LIF overexpression in TgLC mice led to impaired liver functions (Fig. 1N), it is possible that LIF overexpression disrupts proper lipid metabolism in the liver. Here, we compared the levels of lipid metabolites in the liver tissues from TgLC and TgL mice with TAM injection under both fed and fasted conditions. Under the fed condition, LIF overexpression in TgLC mice with TAM injection led to a similar change in the TG levels in the liver tissues as that observed in the serum; TAM injection in TgLC mice resulted in a significant decrease in the levels of the majority of small LC-TGs, which indicated reduced lipogenesis, and a significant increase in the levels of larger LC-TGs (Fig. 2D). The levels of TGs were largely comparable in the liver tissues of TgLC mice with TAM injection under both fed and fasted conditions (Fig. 2D). Similar observations were made in the liver tissues from Balb/c mice bearing C26 or C26-LIF KO tumors (Fig. 2E). Under both fed and fasted conditions, compared with control mice without tumors, mice bearing C26 tumors but not C26-LIF KO tumors at a similar size exhibited a significant decrease in the levels of TGs, especially small LC-TGs, in the liver (Fig. 2E).

To directly examine the effect of LIF on hepatic de novo lipogenesis, we employed deuterated water (D₂O) tracing experiments to label de novo synthesized FA in vivo[26]. TgLC and TgL mice with TAM injections were provided with drinking water containing 20% D₂O for 7 days before the liver tissues were collected for lipid metabolomics analysis. LIF overexpression in TgLC mice significantly decreased hepatic de novo lipogenesis as reflected by the decreased levels of de novo synthesized C16:0, and C18:0, which are the most common and abundant forms of SFAs (Fig. 2F). A very similar decrease in hepatic de novo lipogenesis was observed in the mice bearing C26 tumors but not C26-LIF KO tumors at similar sizes, suggesting that tumor-secreted LIF inhibits hepatic de novo lipogenesis in mice (Fig. 2G). Taken together, these results demonstrate that LIF overexpression in mice inhibits hepatic de novo lipogenesis, which may contribute to the decreased levels of small LC-TGs in the serum and liver tissues in TgLC mice with TAM injection and C26 tumor-bearing mice that develop cachexia.

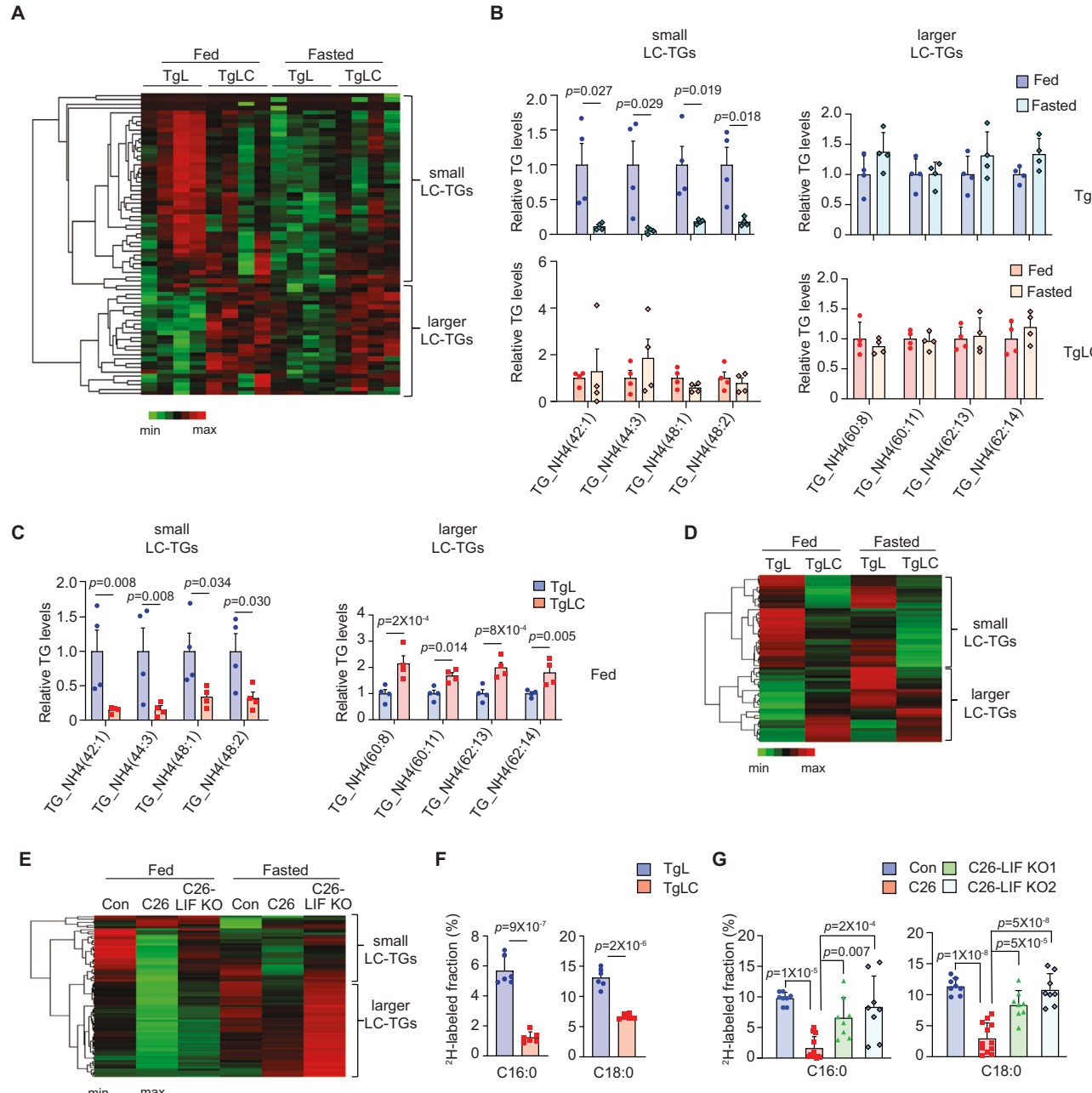

**Fig. 2 | LIF overexpression disrupts lipid homeostasis and decreases hepatic de novo lipogenesis. A** Heatmap of serum TG levels under fed and fasted conditions in TgL and TgLC mice with small LC-TGs (C ≤ 54) and larger LC-TGs (C > 54) clustered together (n = 4/group). **B** Representative serum TG levels in TgL (upper panels) and TgLC (bottom panels) mice injected with TAM under fed and fasted conditions (n = 4/group). The TG levels under the fed condition are designated as 1. **C** Representative serum TG levels under the fed condition (n = 4/group). The TG levels in TgL mice under the fed condition are designated as 1. **D** Heatmap showing the average TG levels in the livers of TgL and TgLC mice injected with TAM under fed and fasted conditions (n = 8/group). **E** Heatmap of the average TG levels in the livers from Balb/c mice with or without C26 or C26-LIF KO tumors (n = 6 for control

(Con) mice without tumors, n = 10 for C26 tumor-bearing mice and n = 6 for C26-LIF KO tumor-bearing mice). **F, G** Hepatic de novo lipogenesis in TgL and TgLC mice with TAM injection (**F**; n = 6/group) and in non-tumor bearing Balb/c mice (n = 8) and Balb/c mice bearing C26 (n = 12) or C26-LIF KO tumors (n = 8) (**G**). Mice drank water containing 20% $D_2O$ for 7 days before tissue collection. Levels of C16:0 and C18:0 in each group were shown. Data are presented as mean ± SD. Each dot represents an individual biological repeat. Both female and male mice were used. For **B, C, F** Two-tailed Student's t-test was applied for comparison between two groups; for **G** One-way ANOVA followed by t-test with Tukey's multiple comparison adjustment was applied for comparison among multiple groups. Source data are provided as Source Data file.

## Blocking the LIF signaling in the liver significantly attenuates cachexia

LIF functions through binding to its receptor complex, which is composed of LIFR and gp130[9]. To investigate the contribution of LIF-induced functional and metabolic changes in the liver to cachexia, we generated a mouse line with a conditional LIF knock-in allele and a

conditional LIFR knockout allele (TgL/LIFR[flox/flox]) (Fig. 3A, B). Liver-specific LIF expression and LIFR knockout were induced in TgL/LIFR[flox/flox] mice by hydrodynamic tail vein injection of Ad5CMVCre-eGFP (Ad-Cre) (Fig. 3A, B). TgL mice with Ad-Cre injection that induces liver-specific LIF expression without LIFR knockout served as controls. Ad-Cre injection induced LIF expression to a

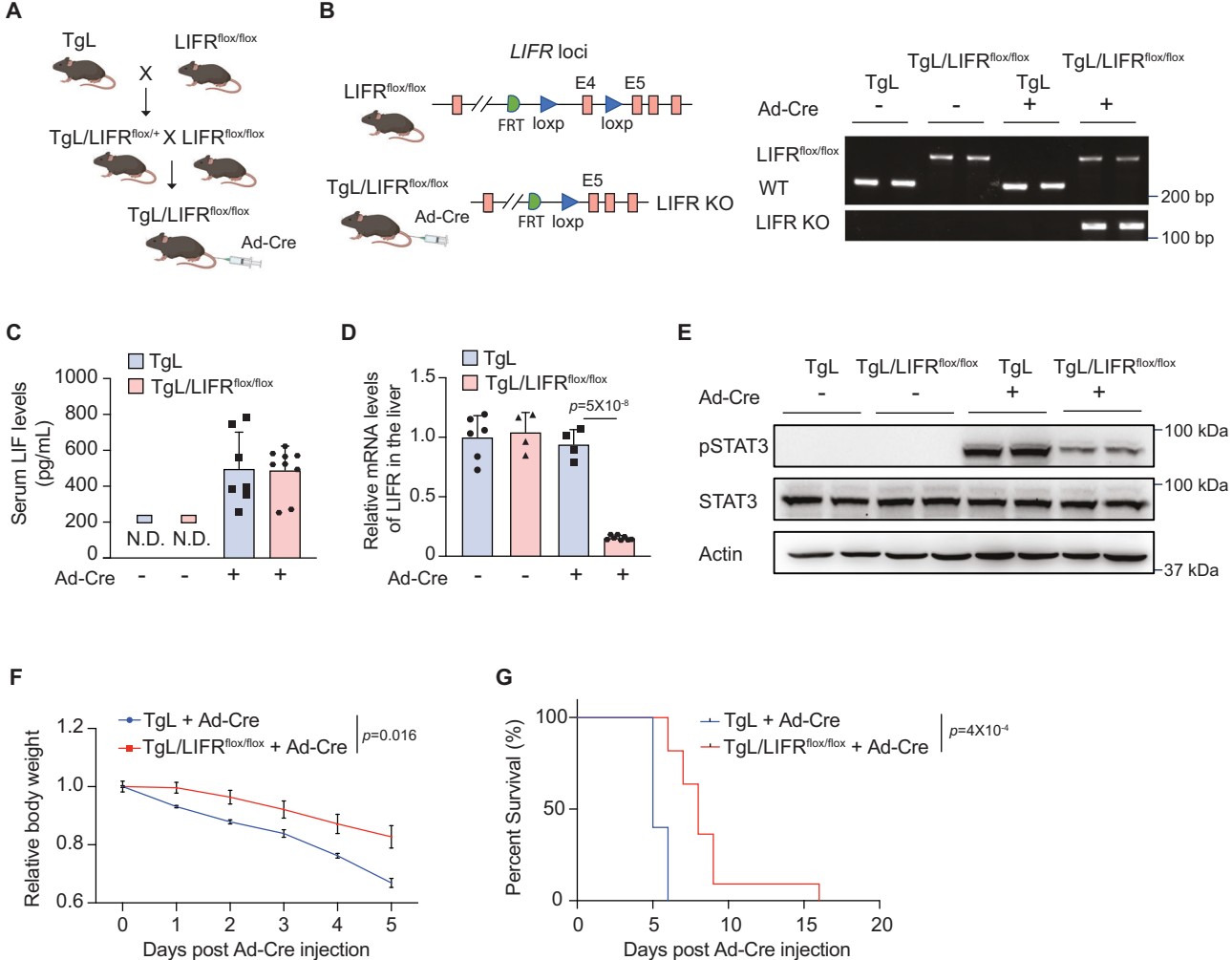

**Fig. 3 | Liver-specific LIFR blockade attenuates cachexia induced by LIF over-expression. A** The generation of TgL/LIFR^flox/flox mice. Liver-specific LIF expression and LIFR knockout was induced in TgL/LIFR^flox/flox mice by hydrodynamic tail vein injection of Ad5CMVCre-eGFP (Ad-Cre). **B** The genotyping analysis of TgL and TgL/LIFR^flox/flox mice with or without Ad-Cre injection by PCR. All mice with the same genotype have similar results. **C** Serum LIF levels in TgL and TgL/LIFR^flox/flox mice with or without Ad-Cre injection (n = 7–9/group). **D** Relative LIFR mRNA levels in TgL and TgL/LIFR^flox/flox mice with or without Ad-Cre injection (n = 4–8/group). **E** The levels of Tyr 705 phosphorylated STAT3 (pSTAT3) and total STAT3 protein in the liver of TgL and TgL/LIFR^flox/flox mice with or without Ad-Cre injection

determined by Western-blot assays. At least three independent biological replicates were performed. **F** Relative body weight of TgL (n = 5) and TgL/LIFR^flox/flox (n = 10) mice post Ad-Cre injection. **G** Kaplan-Meier survival curves of TgL and TgL/LIFR^flox/flox mice post Ad-Cre injection. The day of Ad-Cre injection was denoted as D0. Data are presented as mean ± SD for (**C, D**), and as mean ± SEM for (**F**). N.D. non-detectable. Each dot represents an individual biological repeat. Both female and male mice were used. For **D**: one-way ANOVA followed by t-test with Tukey's multiple comparison adjustment; for **F**: two-way ANOVA followed by Sidak's multiple comparison test; for **G**: two-tailed Kaplan-Meier survival analysis. Source data are provided as Source Data file.

comparable level in the liver of TgL and TgL/LIFR^flox/flox mice as determined by ELISA assays, and Ad-Cre injection significantly decreased LIFR mRNA levels in the liver of TgL/LIFR^flox/flox but not TgL mice as determined by quantitative real-time PCR (qPCR) assays (Fig. 3C, D). STAT3 is a major downstream signaling pathway of LIF, and the levels of STAT3 phosphorylation at Tyr705 (pSTAT3) can reflect STAT3 activity[11,12,14,18]. Ad-Cre injection in TgL mice greatly enhanced the activity of STAT3 in the liver tissues as reflected by the increased levels of pSTAT3 measured by Western-blot assays, whereas Ad-Cre injection in TgL/LIFR^flox/flox mice led to only a very limited increase of pSTAT3 levels in the liver tissue (Figs. 3E and S4). Ad-Cre injection quickly induced cachexia in TgL mice, which exhibited body weight loss and short survival (median survival of 5 days) (Fig. 3F, G). Notably, TgL/LIFR^flox/flox mice with Ad-Cre injection that have liver-specific LIFR knockout to block the LIF signaling exhibited a less pronounced body weight loss and prolonged survival compared with TgL mice with Ad-Cre injection (Fig. 3F, G). These results demonstrate that blocking the LIF signaling in the liver by liver-specific LIFR knockout significantly

attenuates LIF-induced cachexia, suggesting that LIF-induced functional changes in the liver, including its metabolic changes, contribute to LIF-induced cachexia.

## LIF overexpression reduces the expression of lipogenesis genes in TgLC mice

To investigate the underlying mechanism by which LIF overexpression impairs hepatic de novo lipogenesis, we performed transcriptome analysis using RNA-seq assays to compare the gene expression profiles of the liver tissues of TgLC and TgL mice with TAM injection. Among a total of 23,594 genes examined, there were 2271 differentially expressed genes (DEGs) between the liver tissues of TgLC and TgL mice with TAM injection, with 1213 genes being upregulated, and 1058 genes being downregulated (Fig. 4A). Kyoto Encyclopedia of Genes and Genomes (KEGG) pathway enrichment analysis of the DEGs revealed that the top-ranked pathway was metabolic pathways (Fig. 4B), which aligns with the results from the metabolomics analysis showing changes in the levels of many metabolites in the liver tissues

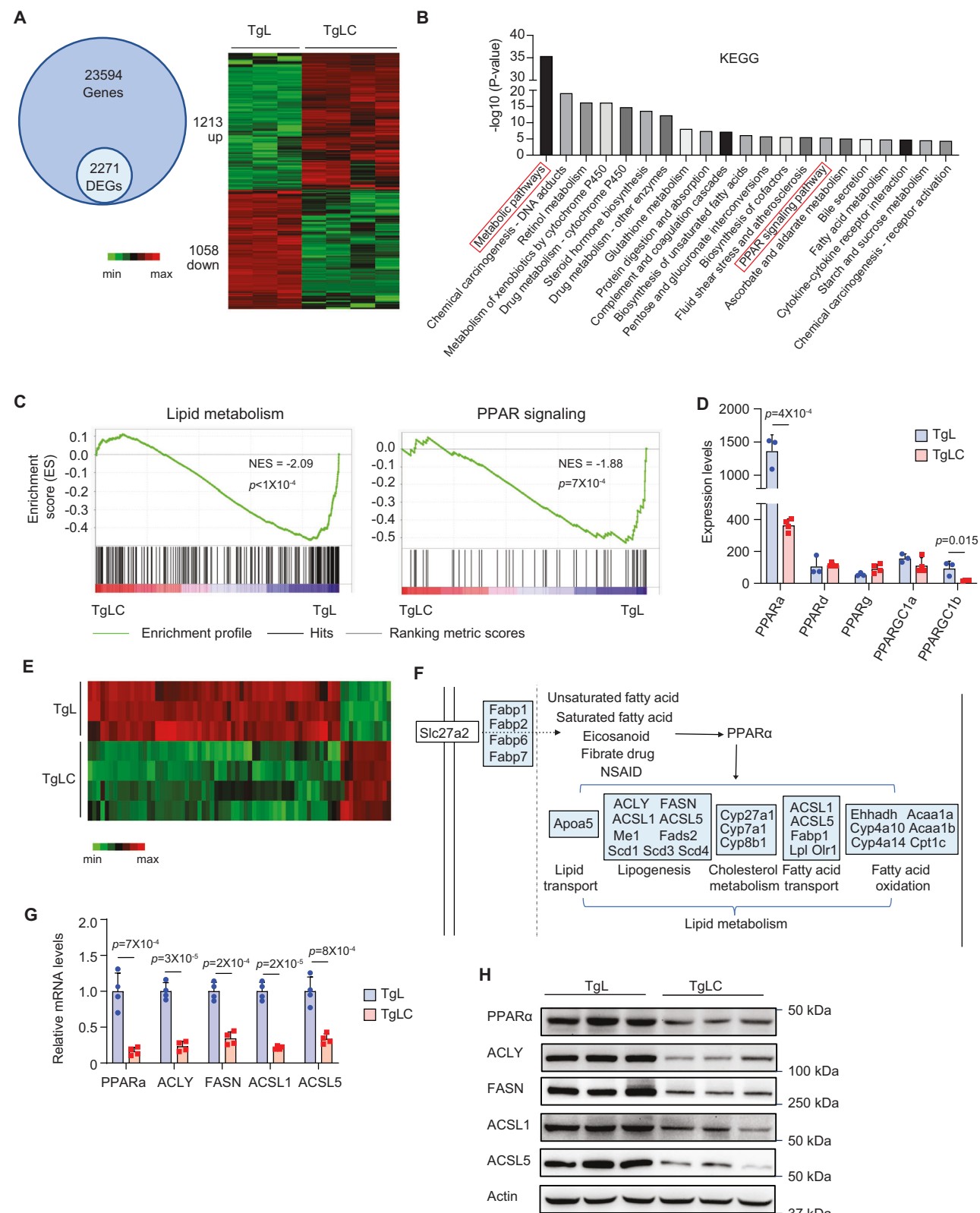

of TgLC mice with TAM injection. PPARs are nuclear receptors that function as transcription factors regulating the expression of genes involved in lipid metabolism[27,28]. KEGG analysis showed that many of the DEGs were enriched in the PPAR signaling pathway (Fig. 4B). Gene-set enrichment analysis (GSEA) revealed that both lipid metabolism and PPAR signatures were downregulated in the livers of TgLC mice with TAM injection (Fig. 4C). The results of the RNA-seq assays showed

that among the 5 PPAR family member genes (*PPARa, PPARd, PPARg, PPARGC1a* and *PPARGC1b*), the mRNA levels of *PPARa* and *PPARGC1b* were significantly reduced in the liver tissues of TgLC mice with TAM injection compared with TgL mice with TAM injection (Fig. 4D). *PPARa* is the most abundant of the PPAR family that is expressed in the liver; its basal expression levels in liver tissues were much higher than those of other PPAR family members (Fig. 4D). The qPCR results showed that

**Fig. 4 | LIF overexpression downregulates the PPARα signaling pathway.**
**A** RNA-seq results showing gene expression levels in the liver from TAM-injected TgLC ($n$ = 4) and TAM-injected TgL mice ($n$ = 3). The number of identified genes and DEGs were shown (left). The DEGs were shown in the Heatmap (right). **B** KEGG analysis of DEGs by the DAVID database. **C** GSEA enrichment plots for lipid metabolism (left) and PPAR signaling pathway (right). **D** RNA-seq results showing the expression levels of genes (*PPARa*, *PPARd*, *PPARg*, *PPARGC1a* and *PPARGC1b*) encoding for five PPAR family members ($n$ = 3 for TgL group, $n$ = 4 for TgLC group). **E** Heatmap of PPARα target genes among the DEGs in the liver from TgLC and TgL mice with TAM injection. **F** KEGG map of PPARα signaling pathway. DEGs were mapped to the "PPARα signaling pathway", according to the "PPARs signaling pathway" map in KEGG with some modifications. DEGs are colored in blue. **G, H** Validation of expression levels of *PPARa* and some of its target genes after LIF overexpression by qPCR assays (**G**; $n$ = 4/group) and Western-blot assays (**H**). At least three independent biological replicates were performed. All data are presented as mean ± SD. Each dot represents an individual mouse. Both female and male mice were used. For **D, G**: two-tailed Student's *t*-test. Source data are provided as Source Data file.

the expression of *PPARa* is most abundant in mouse liver tissues compared with its expression in other tissues, including the heart, muscle, and spleen (Fig. S5A). *PPARa* encodes for PPARα, a master transcription factor for several genes involved in lipogenesis, including *ACLY*, *FASN*, *ACSL1*, and *ACSL5*[29,30]. The RNA-seq data showed differential expression of 72 PPARα targets in the liver tissues of TgLC mice with TAM injection compared with TgL mice with TAM injection (Fig. 4E). Among them, 25 DEGs mapped to the PPARα signaling pathway are involved in lipid metabolism, including 9 genes (*ACLY*, *FASN*, *ACSL1*, *ACSL5*, *ME1*, *FADS2*, *SCD1*, *SCD3* and *SCD4*) involved in lipogenesis (Fig. 4F). The expression changes of PPARα and 4 target genes (*ACLY*, *FASN*, *ACSL1*, *ACSL5*) were validated at both mRNA and protein levels by qPCR and Western-blot assays, respectively (Fig. 4G, H). The decrease in PPARα and ACLY levels was predominantly observed in hepatocytes in the livers of TgLC mice with TAM injection as examined using immunohistochemistry (IHC) assays (Fig. S5B). No obvious changes in the *PPARa* mRNA levels were observed in the livers of the pair-fed TgL mice with TAM injection, indicating that the decrease of PPARα is not due to reduced food intake (Fig. S5C). Consistently, a significant decrease in the mRNA levels of *PPARa*, *ACLY*, *FASN*, *ACSL1* and *ACSL5* was observed in the livers of C26 tumor-bearing mice compared with control mice (Fig. S5D). Notably, little to no change in the mRNA levels of these genes was observed in the livers of C26-LIF KO tumor-bearing mice (Fig. S5D). These results reveal that LIF overexpression decreases the expression levels of *PPARa* and its target genes, especially those involved in lipogenesis in liver tissue, which may lead to decreased hepatic lipogenesis in TgLC mice with TAM injection and C26 tumor-bearing mice.

### LIF overexpression downregulates the expression of *PPARa* via the activation of STAT3 signaling in hepatic cells
LIF exerts its functions through the regulation of various downstream signaling pathways in a highly tissue-, development- and context-specific manner[9]. To investigate the mechanism underlying the downregulation of *PPARa* expression in liver tissue by LIF overexpression, we examined a panel of LIF-regulated downstream pathways, including the STATs, AKT, ERK, and MAPK signaling pathways, in the liver tissues of TgLC and TgL mice with TAM injection. LIF overexpression induced by TAM in TgLC mice clearly enhanced the activity of STAT3 in liver tissues, as reflected by increased levels of pSTAT3, without affecting the total STAT3 protein levels measured by Western-blot and IHC assays (Figs. 5A and S6A). No major changes in the pSTAT3 levels were observed in the liver tissues of the pair-fed TgL mice with TAM injection when compared with the liver tissues of the regular-fed TgL mice with TAM injection (Fig. S6B). Aside from STAT3 signaling, there was no clear activation of other signaling pathways in the livers of TgLC mice with TAM injection (Fig. 5A). Similarly, recombinant mouse LIF protein (rLIF) treatment increased pSTAT3 but not total STAT3 protein levels in primary cultured mouse hepatic cells isolated from wild-type C57BL6/J mice (Fig. 5B). rLIF treatment also markedly decreased the expression of *PPARa* in primary mouse hepatic cells (Fig. 5C). A putative STAT3 binding site was identified in the promoter of the *PPARa* gene (Fig. 5D). Employing chromatin immunoprecipitation (ChIP) assays, we found that the anti-

STAT3 antibody can immunoprecipitate chromatin fragments corresponding to the potential STAT3 binding site in the *PPARa* promoter in hepatic cells treated with rLIF, but not in cells without rLIF treatment, indicating that LIF activates STAT3 to promote its binding to the *PPARa* promoter (Fig. 5E). To further investigate whether LIF-induced STAT3 activation mediates the downregulation of *PPARa* by LIF in hepatic cells, we examined the effect of rLIF on *PPARa* expression in the primary hepatic cells treated with small molecule inhibitors specific for STAT3 and siRNAs targeting STAT3, respectively, to block STAT3 signaling. The downregulation of *PPARa* expression by rLIF in primary hepatic cells was largely abolished by two small molecule STAT3 inhibitors, Stattic and Galiellalactone, as well as by two siRNA oligos targeting STAT3 (Fig. 5F, G). Collectively, these results indicate that LIF overexpression downregulates *PPARa* expression in hepatic cells, mainly through the activation of STAT3 signaling.

### Activating PPARα by fenofibrate restores lipid homeostasis in the liver and inhibits cachexia
Our results demonstrate that LIF overexpression downregulates PPARα expression and its target genes involved in lipogenesis, which suggest that LIF overexpression may decrease lipogenesis and disrupt lipid homeostasis in the liver, thereby inducing cachexia. To test this hypothesis, we employed fenofibrate, a fibric acid derivative widely used as a PPARα agonist[31], to investigate whether PPARα activation ameliorates the disrupted lipid metabolism induced by LIF overexpression. While LIF overexpression decreased the expression of *ACLY*, *FASN*, *ACSL1*, and *ACSL5* in the liver tissues of TgLC mice injected with TAM, their mRNA and protein levels were significantly higher in TgLC mice fed with a fenofibrate-containing diet starting 3 days before TAM injection (Fig. 6A, B). In contrast, in TgL mice with TAM injection, fenofibrate diet did not significantly impact the expression of these genes in the liver tissues where *PPARa* levels were high (Fig. S7A). Similarly, while the expression levels of *ACLY*, *FASN*, *ACSL1*, and *ACSL5* in the liver tissues were decreased in C26 tumor-bearing mice compared with control mice without tumors, their expression levels were significantly increased in the liver tissues of the mice fed with a fenofibrate diet (Fig. 6C, D). These results clearly show that LIF overexpression downregulates the expression of PPARα and its downstream targets in the liver, and that the downregulation of PPARα target genes by LIF overexpression can be blocked by fenofibrate.

We then investigated whether fenofibrate can restore hepatic lipid homeostasis in TgLC mice with LIF overexpression. While TAM injection in TgLC mice led to lower levels of small LC-TGs and higher levels of larger LC-TGs in the liver, fenofibrate significantly increased the levels of the majority of small LC-TGs, and decreased the levels of many larger LC-TGs in the liver of TAM-injected TgLC mice (Figs. 6E and S7B). A similar effect of fenofibrate on lipid metabolites was observed in C26 tumor-bearing mice; compared with the livers of C26 tumor-bearing mice under the fed condition, which had a majority of small LC-TGs at lower levels than control mice, fenofibrate significantly increased the levels of many small LC-TGs in the livers of C26 tumor-bearing mice (Fig. 6E).

To further investigate the contribution of impaired lipid metabolism in the liver to cachexia induced by LIF overexpression, we tested whether fenofibrate inhibits cachexia development in both

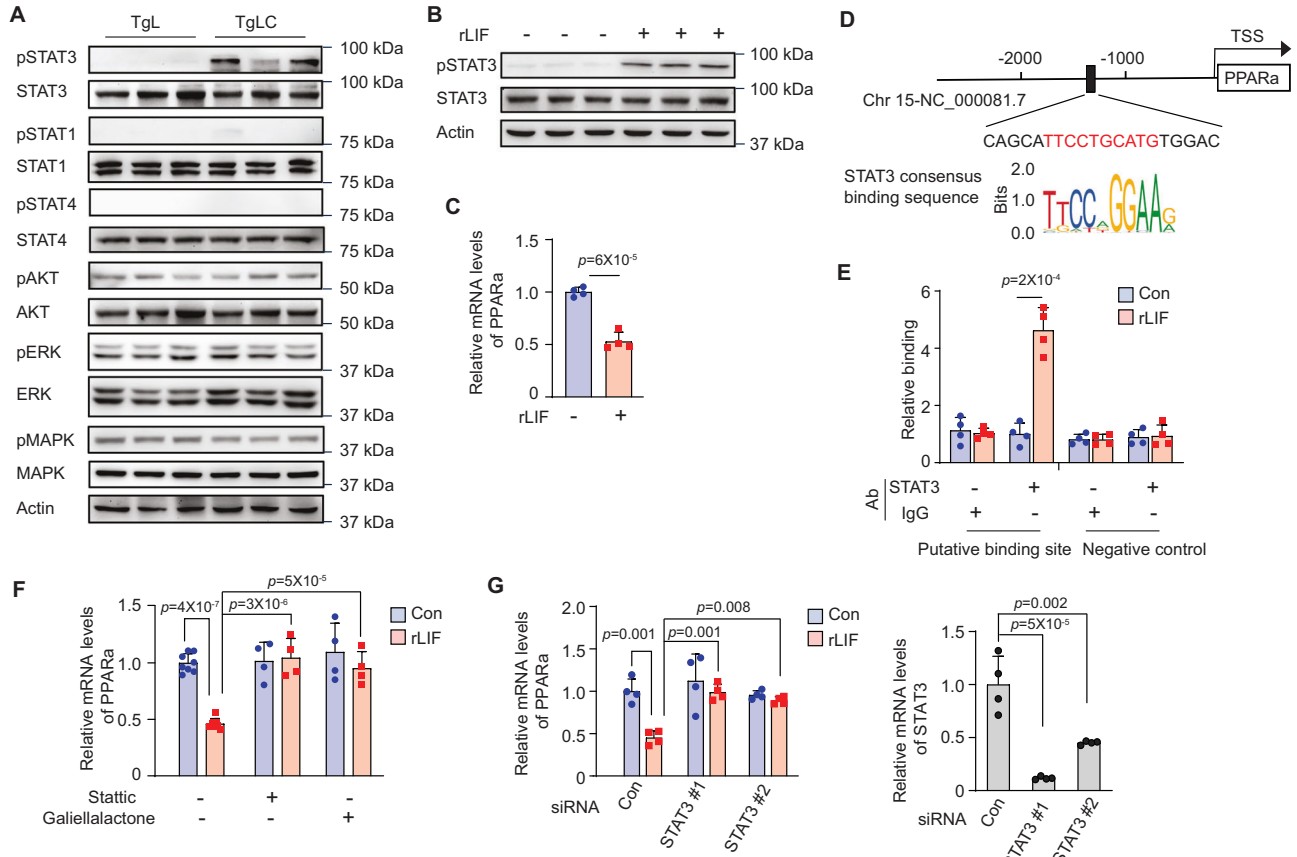

**Fig. 5 | LIF downregulates the expression of *PPARa* via the activation of the STAT3 signaling in hepatic cells. A** LIF overexpression increased the levels of pSTAT3 but not total STAT3 protein or other LIF downstream pathways, including STAT1, STAT4, AKT, ERK, and MAPK in the liver of TgLC mice post TAM injection as determined by Western-blot assays. At least three independent biological replicates were performed. **B** rLIF treatment (100 ng/ml for 30 min) of primary mouse hepatic cells increased the levels of pSTAT3 but not total STAT3 protein as determined by Western-blot assays. Three replicates were presented in each group. At least three independent biological replicates were performed. **C** rLIF treatment decreased the mRNA levels of *PPARa* in primary cultured hepatic cells (*n* = 4/group). **D** The sequence and location of a putative STAT3 binding site in the mouse *PPARa* pro-moter region. TSS: transcription start site. **E** rLIF increased the binding of STAT3 to

a putative STAT3 binding site in the promoter of *PPARa* as determined by ChIP assays in primary mouse hepatic cells. A chromatin region without STAT3 binding site was included as a negative control (*n* = 4/group). **F** Blocking STAT3 by STAT3 inhibitors, Stattic (2 μM) or Galiellalactone (5 μM), largely abolished the inhibitory effect of rLIF on the expression of *PPARa* in primary mouse hepatic cells. The mRNA levels of *PPARa* were determined by qPCR assays and normalized to β-actin (*n* = 4–8/group). **G** STAT3 siRNAs largely abolished the inhibitory effect of rLIF on *PPARa* expression in primary mouse hepatic cells. Left: relative *PPARa* mRNA levels; right: relative STAT3 mRNA levels in primary cultured hepatic cells (*n* = 4/group). All data are presented as mean ± SD. Both female and male mice were used. For **C**, **E**: two-tailed Student's *t* test; for **F**, **G**: one-way ANOVA followed by *t*-test with Tukey's multiple comparison adjustment. Source data are provided as Source Data file.

TgLC and C26 tumor-bearing mice. Notably, fenofibrate significantly reduced the body weight loss of TgLC mice post TAM injection (Fig. 6F). TgLC mice with TAM injection fed with fenofibrate diet showed a trend of improved food intake compared with TgLC mice with TAM injection fed with regular chow (Fig. S7C). Furthermore, fenofibrate largely blocked lean mass loss and also exhibited a trend of less extensive fat mass loss, although the difference in fat mass loss was not significant (Fig. 6G). Importantly, fenofibrate prolonged the sur-vival of TgLC mice with TAM injection (Fig. 6H). Similar results were obtained in C26 tumor-bearing mice; fenofibrate significantly delayed body weight loss and prolonged the survival of C26 tumor-bearing mice (Fig. 6I, J). Collectively, these results demonstrate that fenofibrate activates PPARα and its downstream targets involved in lipogenesis in the liver to ameliorate the impaired lipid metabolism induced by LIF overexpression in TgLC mice and C26 tumor-bearing mice, which in turn inhibits cachexia development (Fig. 6K).

## Discussion

Cancer cachexia is a metabolic syndrome characterized by unintended weight loss, muscle and fat wasting, which occurs in many advanced

cancer patients, with many progressing to death[1]. Currently, the underlying mechanisms of cancer cachexia are still not well-understood. In addition to the competition of nutrients between tumor and host cells, cytokines and other factors produced by tumor cells and/or cells in the tumor microenvironment play important roles in cachexia[1,5,32,33]. The multi-functional cytokine LIF is involved in many important biological processes. The binding of LIF to LIFR induces its hetero-dimerization with gp130. The formation of this receptor com-plex activates the receptor-associated Janus kinases (JAKs) by phos-phorylating receptor docking sites, which in turn leads to the recruitment and activation of Src Homology-2 (SH2) domain-containing proteins, such as STAT3[34,35]. Recent studies, including our own, have shown that LIF is frequently overexpressed in many cancer types, and elevated serum LIF levels have been observed in patients with different cancer types, including PDACs, OACs, and nasophar-yngeal carcinomas (NPCs)[17–19,36]. Furthermore, LIF overexpression is associated with a poor prognosis in cancer patients, indicating an important role of LIF in tumor progression. LIF overexpression pro-motes cancer cell proliferation, metastasis, immune evasion, stem-ness, and metabolic reprogramming, all of which contribute to

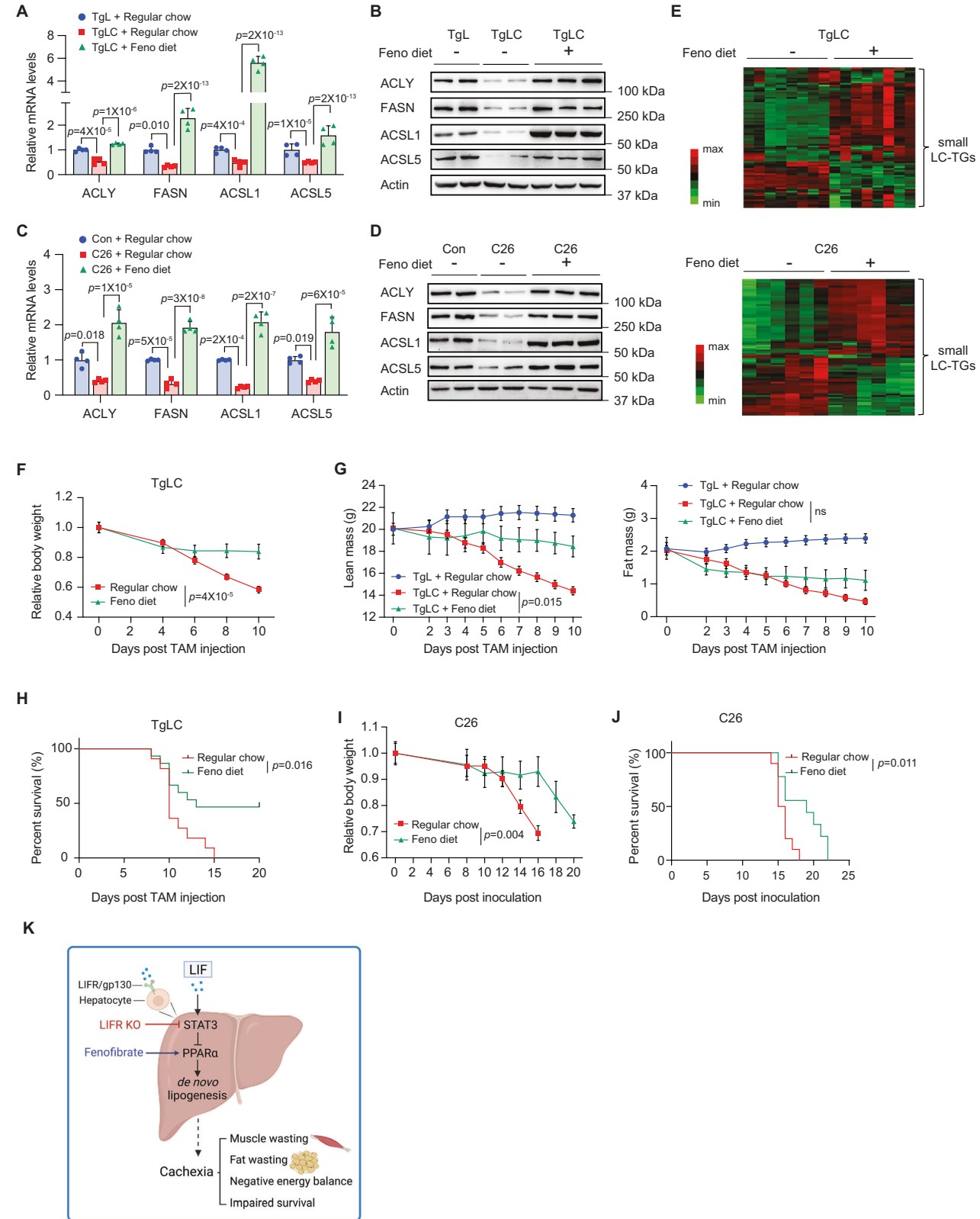

tumorigenesis[10]. Importantly, LIF has been suggested as a cachexia-inducing factor[9,11,12]. The secretion of LIF from several human and mouse cancer cell lines has been associated with cancer cachexia development in mice inoculated with these cancer cells[11,37]. This study observes that C26 tumor-bearing mice exhibit progressive body weight loss and poor survival, which is consistent with previous reports[9,11]. Importantly, LIF knockout in C26 cells significantly

mitigates cachexia in tumor-bearing mice[11,38]. While the connection of LIF with cachexia has been established, the underlying mechanisms whereby LIF promotes cachexia remain unclear. Intrinsic LIF signaling in cancer cells has been shown to promote cell growth and proliferation with increased energy demands, which may enhance the competition of cancer cells with host cells for nutrients[9,39]. At the same time, increased LIF production from tumors may have systemic effects on

**Fig. 6 | Fenofibrate restores hepatic lipid homeostasis and inhibits cachexia induced by LIF overexpression. A, B** mRNA levels (**A**; $n = 4$/group) and protein levels (**B**) of PPARα target genes in the livers of TgLC mice fed with regular chow or fenofibrate (Feno) diet (0.2% w/w). **C, D** mRNA levels (**C**; $n = 4$/group) and protein levels (**D**) in the liver of Balb/c mice bearing with or without C26 tumors fed with regular chow or fenofibrate diet. **E** Fenofibrate diet increased the levels of majority of small LC-TGs (C ≤ 54) in the liver of TgLC mice and C26 tumor-bearing mice. Heatmap showing the TG levels in the liver of TgLC mice ($n = 8$/group) and C26 tumor-bearing mice (n = 6/group) fed with regular chow or fenofibrate diet. **F** Body weight of TgLC mice fed with regular chow or fenofibrate diet ($n = 5$/group). **G** Lean mass (left) and fat mass (right) of TgLC mice fed with regular chow or fenofibrate diet post TAM injection. $n = 6–8$/group. **H** Kaplan-Meier survival curves of TgLC mice fed with regular chow or fenofibrate diet. **I** Body weight of C26 tumor-bearing mice fed with regular chow or fenofibrate diet. The day of C26 cells inoculation was denoted as D0. $n = 5$/group. **J** Kaplan-Meier survival curves of C26 tumor-bearing mice. **K** The diagram depicting the mechanism by which LIF induces cachexia. The diagram was created with BioRender.com. Data are presented as mean ± SD for (**A**, **C**), and as mean ± SEM for (**F**, **G**, **I**). Each dot represents an individual mouse. Both female and male mice were used. ns: non-significant. For **A**, **C**: one-way ANOVA followed by *t*-test with Tukey's multiple comparison adjustment; for **F**, **G**, **I**: two-way ANOVA followed by Sidak's multiple comparison test; for **H**, **J**: two-tailed Kaplan-Meier survival analysis. Source data are provided as Source Data file.

the host, contributing to cachexia. However, most cancer cachexia mouse models are unable to distinguish the contribution of LIF to cachexia through cell-intrinsic mechanisms or the systemic effects to the host.

In this study, we established an inducible transgenic LIF over-expression mouse model (TgLC) which allows us to characterize the effect of systemic LIF elevation on cachexia. Our findings revealed that the systemic elevation of LIF levels in TgLC mice significantly reduced food intake, and caused a shift of metabolic fuel from carbohydrates to fat in these mice. Although LIF overexpression also reduced TEE, col-lectively, TgLC mice with LIF overexpression exhibited a negative energy balance and developed cachexia syndrome displaying muscle mass loss, adipose tissue loss, and impaired survival (Fig. 6K).

As a metabolic syndrome, cachexia is caused by profound meta-bolic alterations. In TgLC mice with LIF overexpression, disrupted lipid homeostasis was observed during cachexia development. The liver is an important organ that controls systemic metabolism. While limited evidence suggests the potential role of the liver in cachexia develop-ment, including its participation in acute-phase response, and Cori cycle gluconeogenesis[1,4,5], its precise role in cachexia and underlying mechanisms are largely unexplored. Results from this study showed that hepatic de novo lipogenesis was significantly inhibited during cachexia development in TAM-injected TgLC mice with LIF over-expression. In turn, the levels of small LC-TGs mostly containing SFAs were significantly reduced in the liver tissues in TAM-injected TgLC mice with LIF overexpression. Meanwhile, the levels of larger LC-TGs mostly containing PUFAs, reflecting the ongoing lipolysis, were greatly increased in the liver tissue. The serum of TgLC mice with LIF over-expression exhibited a very similar change in TG levels. It is worth noting that there are inconsistent reports regarding the changes in TG levels during cachexia development[40–43], which may be due to the measurement of the mixed pool of TGs. Importantly, results from this study suggest that LIF-induced functional changes in the liver, including its metabolic changes, contribute to LIF-induced cachexia; blocking the LIF signaling in the liver by liver-specific LIFR knockout partially abolished LIF overexpression-induced cachexia in mice.

Mechanistically, LIF overexpression inhibited hepatic de novo lipogenesis through the STAT3/PPARα axis. LIF overexpression activated STAT3 signaling, which led to the downregulation of *PPARα* expression in hepatocytes both in vitro and in vivo. In turn, the expression of a group of PPARα target genes involved in lipo-genesis, including *ACLY*, *FASN*, *ACSL1*, and *ACSL5*, was significantly reduced. Activating PPARα by the fenofibrate diet significantly increased the expression of PPARα target genes, restored lipid homeostasis in the liver, and more importantly, significantly inhib-ited LIF overexpression-induced cachexia. The connection between STAT3 signaling and cachexia has been reported previously. It has been reported that the activation of STAT3 signaling supports lipase ATGL and its co-activator CGI-58 dependent adipocyte lipolysis and increases serum leptin levels that may influence cachexia-associated anorexia[44,45]. Results from our study reveal an important role of the STAT3 signaling in mediating LIF-induced impairment of hepatic de novo lipogenesis.

Interestingly, a previous study on cancer cachexia using a non-small cell lung cancer mouse model reported decreased PPARα nuclear localization and PPARα-dependent ketogenesis in the liver of mice that developed cancer cachexia. These mice exhibited hypoke-tonemia with decreased serum levels of β-hydroxybutyrate (BHB), the most abundant form of ketone body[46]. It is worth noting that in TAM-injected TgLC mice with LIF overexpression, no significant change in BHB levels was observed during cachexia development (Fig. S8), indicating that LIF does not play an obvious role in ketogenesis during cachexia development. LIF is often highly expressed in different types of cancers, and the mechanisms of LIF overexpression in cancers are not completely understood, as LIF can be transcriptionally regulated by many different factors, including HIF-2α, TGF-β, STAT5, and p53, in a highly context-dependent manner[10]. Future studies are needed to better understand the overexpression of LIF in cancers and as well as the contribution of these LIF regulators to cachexia. Since many dif-ferent factors can contribute to cachexia, future studies are needed to validate the relevance of our findings in cancer mouse models and human patients, especially those cancers with LIF overexpression. In addition, cachexia is a syndrome affecting multiple organs including muscle, fat, brain, and liver tissues[6]. A very recent study showed that the obesity-associated LIF receptor (LIFR)/STAT3 signaling in adipo-cytes can modulate lipid metabolism in the liver and contribute to liver triglyceride (TG) accumulation[47], which highlights the importance of the cytokine-induced inter-tissue crosstalk in metabolic dysregulation and cachexia development. While this study reveals the LIF-induced functional and metabolic changes in the liver and its contribution to cachexia, further studies are needed to understand the role of LIF in additional organs, the signaling pathway(s) that mediate the role of LIF, and its contribution to cachexia.

In summary, we established a transgenic LIF overexpression mouse model that robustly induces cachexia. This study demonstrates the systemic effect of LIF on cachexia, and reveals that LIF over-expression disrupts hepatic de novo lipogenesis via the STAT3/PPARα axis as an important underlying mechanism. Blocking the LIF signaling in the liver or re-activating PPARα inhibits LIF overexpression-induced cachexia in mice. This study provides mechanistic insights into cachexia and suggests restoring PPARα-dependent hepatic de novo lipogenesis as a potential strategy to treat cachexia.

## Methods

### Mice

All animal experiments were approved by the Institutional Animal Care and Use Committee of Rutgers University. As cachexia occurs in both males and females, both males and female mice were used. Wild type C57BL6/J mice, Balb/c mice, and R26-CreERT2 mice (Stock No: 008463) were obtained from the Jackson Laboratory. LIFR^flox/flox mice were obtained from The European Mouse Mutant Archive (EM: 09032). TgL mice were generated at Rutgers Transgenic Mouse Facility. Mice were housed under a 12-hour light/dark cycle with 6 am light on and 6 pm light off. The temperature was maintained between 70° and 74 °F and the humidity was between 30 and 70%. The sequences of primers for PCR genotyping of TgL mice are listed in Supplementary Table 2. TgLC

mice were established by crossing TgL mice with R26-Cre[ERT2] mice. The overexpression of LIF in TgLC mice was induced by injection of TAM (32 μg/g of body weight for female mice and 64 μg/g of body weight for male mice; *i.p.*, once). TgL/LIFR[flox/flox] mice were generated by crossing LIFR[flox/flox] mice with TgL mice. The hepatocyte-specific LIFR deletion was induced in TgL/LIFR[flox/flox] mice by a one-time tail vein injection of Ad5CMVCre-eGFP virus (2x10E9 pfu/mouse, UI Viral Vector Core). Pair-fed TgL mice were provided with the amount of food that matched that consumed by TgLC mice with TAM injection. C26 murine colon carcinoma cells (Cell lines service, Cata# 400156) and C26-LIF KO cells with LIF knockout were used to form syngeneic xenograft tumors. The tumor sizes were not exceeded the maximal tumor size (2000 mm³) permitted by the Institutional Animal Care and Use Committee of Rutgers University. C26-LIF KO cells were obtained by knocking out LIF in C26 cells using the CRISPR/Cas9 system as described previously[48]. The sequences of sgRNAs targeting LIF are listed in Supplementary Table 2. Cells used are not in the misidentified lines list and were regularly tested for mycoplasma using the Lookout Mycoplasma PCR detection kit (Sigma) to ensure the absence of mycoplasma. Eight to ten-week-old Balb/c mice were inoculated (*s.c.*) with C26 or C26-LIF KO cells to form syngeneic xenograft tumors.

Mice were fed with a regular chow diet (PicoLab Mouse Diet 20 5053, Lab diet) with or without fenofibrate (0.2% w/w, Cayman chemical). Fenofibrate diets were custom-made by TestDiet. For fenofibrate diet experiments, mice were switched to fenofibrate diet 3 days prior to TAM injection. The investigators were blinded to the group allocation during experiments and when assessing outcomes.

## D₂O labeling

Mice were provided with drinking water containing 20% $D_2O$ (Cambridge Isotope Laboratories) for 7 days before tissue collection. The lipids from livers were extracted and saponified before the LC-MS analysis[49]. The liver samples were pulverized and mixed with pre-cooled methanol (12 μL/mg of tissue). The mixed samples were added with −20 °C methyl tert-butyl ether (MTBE, 40 μL/mg of tissue). After shaking the samples for 6 min at 4 °C, $H_2O$ (10 μL/mg of tissue) was added followed by centrifugation for 2 min. The top MTBE layer was transferred and air-dried. Subsequently, the samples were re-suspended in 1 mL of saponification solvent (0.3 mol/L KOH in 90:10 methanol/$H_2O$), and incubated at 80 °C for 1 h. After incubation, the samples were put on ice for 3 min, followed by the addition of 100 μL of formic acid and 300 μL of hexanes, which resulted in two layers after vortexing the samples. The top layer was transferred to a new tube. This step was repeated to obtain a final volume of 600 μL. The extracted samples were air-dried and then re-suspended in 150 μL of resuspension solvent (50:50 isopropanol/methanol) followed by centrifugation at 4 °C for 10 min. The resulting supernatant was transferred to LC-MS vials for further analysis. The deuterium labeling was calculated after isotope natural abundance correction using AccuCor[50].

## Metabolic cages and body composition analyses

Mice were individually housed in the Promethion Metabolic Cages system (Sable system) under a 12-hour light-dark cycle for 7 days. During this period, food and water intake, $O_2$ consumption, $CO_2$ production, and spontaneous activity were measured. Raw data were collected by the Promethion system and processed by the Promethion software package using the Macro 13 function, which produced standardized output formats for the metabolic variables of interest at each cage. The processed data generated by Macro 13 were then analyzed by the CalR software (https://calrapp.org) as described previously[21]. The mouse body composition analysis (fat and lean mass) was conducted by the EchoMRI™−100H body composition analyzer according to the manufacturer's instruction.

## Serum biochemistry analysis

The blood serum samples were analyzed by the Element DC5X™ Veterinary Chemistry Analyzer (Hesk) performed at Rutgers In Vivo Research Services (IVRS) core facility. Biochemistry parameters examined included BUN, albumin, ALP, and GGT.

## Metabolites extraction and metabolomics analysis

Polar metabolites were extracted as described previously[51–53]. Briefly, polar metabolites were extracted from serum using the extraction buffer containing methanol: acetonitrile: $H_2O$ (40:40:20). The metabolites were analyzed using a Vanquish Horizon UHPLC system (Thermo Fisher Scientific) with an XBridge BEH Amide column (150 mm × 2.1 mm, 2.5 μm particle size, Waters). The solvent and run conditions for UHPLC were described previously[54]. MS scans were obtained in both negative and positive ion modes with a resolution of 70,000 at m/z 200, in addition to an automatic gain control target of $3 \times 10^6$ and m/z scan range of 72 to 1000. Metabolite data were obtained using the MAVEN software package[55] (mass accuracy window: 5 ppm).

For lipidomic metabolites extraction, 10 μL serum were mixed with 75 μL methanol and 250 μL MTBE, and 10 mg homogenized tissues were mixed with 120 μL methanol and 400 μL MTBE. After shaking the samples for 6 min at 4 °C, $H_2O$ (62 μL for serum and 100 μL for tissues) was added followed by centrifugation for 2 min. Supernatants were dried down for further lipidomics analysis. The reverse phase separation was performed on a Vanquish Horizon UHPLC system with a Poroshell 120 EC-C18 column (150 mm × 2.1 mm, 2.7 μm particle size, Agilent InfinityLab) using a gradient of solvent A (90%:10% $H_2O$:MeOH with 34.2 mM acetic acid, 1 mM ammonium acetate, pH 9.4), and solvent B (75%:25% IPA:methanol with 34.2 mM acetic acid, 1 mM ammonium acetate, pH 9.4). The gradient was 0 min, 25% B; 2 min, 25% B; 5.5 min, 65% B; 12.5 min, 100% B; 19.5 min, 100% B; 20.0 min, 25% B; 30 min, 25% B. The flow rate was 200 μl/min. Injection volume was 5 μL and column temperature was 55 °C. The autosampler temperature was set to 4 °C and the injection volume was 5 μL. The full scan mass spectrometry analysis was performed by using a Thermo Q Exactive PLUS with a HESI source as described previously[54]. The lipid identification was performed using MS-DIAL[56], and the lipid quantitation was obtained using the MAVEN software package[55] (mass accuracy window: 5 ppm).

## Isolation and treatment of primary mouse hepatocytes

Primary mouse hepatocytes were isolated as described previously[57]. Briefly, mice were anesthetized by *i.p.* injection of Ketamine/Xylazine mix. The inferior portal vein was cannulated. The liver was perfused with Krebs-Ringer solution containing EGTA and digested with Krebs-Ringer solution containing Liberase™ (Roche) and $CaCl_2$. Stattic (Sigma), Galiellalactone (R&D Systems) and rLIF (Millipore) were used to treat primary mouse hepatocytes. STAT3 siRNAs (SASI_Mm01_00106320, SASI_Mm01_00106321, Sigma) were used to knock down STAT3 in mouse hepatocytes as described previously[14].

## ChIP assays

ChIP assays were performed as described previously[58]. Primary mouse hepatocytes with or without rLIF treatment were used for ChIP assays. The anti-STAT3 antibody (Santa Cruz, Cata# sc-8019) was used for ChIP assays. The primers were designed to encompass the potential STAT3-binding element in the *PPARa* promoter region. The sequences of primers are listed in Supplementary Table 2.

## ELISA assays

LIF levels in the mouse serum were determined by ELISA assays using a mouse LIF Duoset kit (R&D Systems) according to the manufacturer's instruction.

## H&E and IHC staining assays

Formalin-fixed and paraffin-embedded (FFPE) muscle and WAT tissue sections were stained with hematoxylin and eosin as described previously[58]. IHC staining of FFPE liver tissue sections were performed as described previously[59]. Briefly, tissue sections were deparaffinized in xylene and rehydrated in ethanol and water, followed by antigen retrieval by boiling slides in antigen unmasking solution (Cata#: h3300, Vector Laboratories) for 10 min. The following primary antibodies were used: anti-pSTAT3 (cell signaling, Cata# 9145 S), anti-STAT3 (Santa Cruz, Cata# sc-8019), anti-PPARα (Thermo Fisher Scientific, Cata# MA5-37652) and anti-ACLY (Santa Cruz, Cata# sc-517267). The dilution for all antibodies was 1:10.

## qPCR assays

qPCR assays were performed as described previously[59]. Total RNA was extracted by RNeasy kits (QIAGEN), and cDNA was synthesized using TaqMan™ Reverse Transcription Reagents (Applied Biosystems). qPCR was performed by SYBR Green PCR Master Mix (Roche). The expression of β-actin gene was employed for normalization of the expression levels of analyzed genes. qPCR primers are listed in Supplementary Table 2.

## RNA-seq assays

Total RNA from the liver tissue of TgL and TgLC mice with TAM injection was extracted using RNeasy kits (QIAGEN), and then subjected to RNA-Seq assays. Raw data (raw reads) with the fastq format were processed through the fastp software, then mapped to the mouse reference genome. Differential expression analysis was performed by the DESeq2 method[60]. False discovery rate (FDR) was used to control the multiple comparisons based on the Benjamini and Hochberg method. An FDR cutoff of 0.05 and an absolute fold change of 2 were set as the threshold to select for genes with significantly differential expression.

## Bioinformatic analysis

KEGG pathway analysis was performed by the DAVID database[61] to explore pathways enriched for the DEGs. The DEGs were searched against the PPARα signaling pathway maps. GSEA was performed by the GSEA software[62]. Heatmap was generated using the Cluster 3.0 software and visualized via Treeview as described previously[63,64]. Briefly, raw data were first converted to Log transform data and then used for Hierarchical Clustering analysis to generate the cdt file, which was then visualized with Treeview.

## Western-blot assays

Standard Western-blot assays were used as described previously[59]. The following primary antibodies were used: anti-PPARα (Abcam, Cata# ab24509), anti-ACLY (Santa Cruz, Cata# sc-517267), anti-FASN (Santa Cruz, Cata# sc-48357), anti-ACSL1 (cell signaling, Cata# 4047 S), anti-ACSL5 (Santa Cruz, Cata# sc-365478), anti-STAT3 (Santa Cruz, Cata# sc-8019), anti-pSTAT3 (cell signaling, Cata# 9145 S), anti-STAT1 (cell signaling, Cata# 14994 S), anti-pSTAT1 (cell signaling, Cata# 9177 S), anti-STAT4 (cell signaling, Cata# 2653 S), anti-pSTAT4 (cell signaling, Cata# 4134 S), anti-AKT (Santa Cruz, Cata# sc-5298, 1:2000 dilution), anti-pAKT (cell signaling, Cata# 9018 S), anti-pERK (cell signaling, Cata# 4376), anti-ERK (cell signaling, Cata# 9102), anti-pMAPK (cell signaling, Cata# 4511), anti-MAPK (cell signaling, Cata# 9212) and anti-β-actin (Sigma, Cata# A5441, 1:100,000 dilution) antibodies. Other than anti-AKT and anti-β-actin antibodies, the dilution for all other antibodies was 1:1000.

## Statistical analysis

The data were presented as the mean ± SEM or mean ± SD as indicated in the figure legend. Two-tailed Student's $t$-test was applied for statistical analysis between two groups. One-way ANOVA followed by $t$-test with Tukey's multiple comparison adjustment was applied for statistical analysis among multiple groups. The curves of fat, lean mass and body weight loss were compared by two-way ANOVA followed by $t$-test with Sidak's multiple comparison adjustment. The longitudinal measurements on food intake, TEE, $VO_2$, and $VCO_2$ were analyzed using the generalized estimating equation (GEE) method to account for the within-animal correlation. A linear regression model was specified for each outcome with the group and day interaction as the regression terms while adjusting for animal weight. The within-animal correlation structure was specified using the first-order autoregressive correlation. The analysis was done using the geepack package in R. The detailed results of GEE analysis are presented in the supplementary materials. The mouse survival curves were analyzed by Kaplan-Meier method. $P$-value less than 0.05 was considered statistically significant.

## Reporting summary

Further information on research design is available in the Nature Portfolio Reporting Summary linked to this article.

## Data availability

Source data are provided with this paper. RNA-seq data generated in this study have been deposited in the Gene Expression Omnibus (GEO) database under accession number GSE245198. Metabolomics data have been deposited at the Metabolomics Workbench under Project ID: PR001766. Source data are provided with this paper.

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

## Acknowledgements

This work was supported in part by grants from NIH 1R01CA260837, R01CA260838, New Jersey Commission on Cancer Research (NJCCR) COCR22PRG004, and the Ludwig Princeton Branch of the Ludwig Institute for Cancer Research to W.H., and grants from NIH 1R01CA227912 and 1R01CA214746 to Z.H.. X.Y. and F.Z. are supported by NJCCR Fellowship Award. G.G. is supported by NIH R01GM135258 and Veteran Affairs BX002741. E.W. is supported by NCI 1OT2CA278609-01, CRUK (CGCATF-2021/100022) and the Ludwig Princeton Branch of the Ludwig Institute for Cancer Research. This work was also supported in part by the Rutgers Cancer Institute of New Jersey Metabolomics Shared Resource (NCI-CCSG P30CA072720-5923). We thank Jennifer Hostettler of the Medical Writing Services of Rutgers Cancer Institute of New Jersey for assistance editing this paper.

## Author contributions

X.Y. carried out the experiments, analyzed data and wrote the paper; J.W., C.C., F.Z., J.L., H.X., M.I., and M.G. carried out experiments; G.G. assisted with liver function and PPARα signaling analysis; H.L. assisted with statistical data analysis; W.Z. assisted with the mouse model to induce liver-specific LIF expression and LIFR knockout; F.W. assisted with the isolation and analysis of primary mouse hepatocytes; X.S. assisted with metabolomics analysis; E.W. assisted with the analysis of metabolic phenotyping as well as the body composition of mice; Z.F. and W.H. designed experiments, analyzed data and wrote the paper.

## Competing interests

The authors declare no competing interests.
