## [Peer Review File · Nature Communications]

REVIEWER COMMENTS

Reviewer #1 (Remarks to the Author):

The manuscript entitled: Leukemia inhibitory factor suppresses hepatic de novo lipogenesis and induces cachexia, is a very well constructed update to the importance of LIF to cachexia in response to tumors. The work begins by reproducing earlier studies showing the necessity of LIF to the cachexia produced by the C26 tumor. It adds the important work of showing the sufficiency of LIF to C26-induced cachexia and then takes on the underlying mechanism of the effect of LIF on adipose, the most obvious and deeply impacting tissue in the C26 cachexia disease. The findings of the importance of liver location and subsequent PPAR biology in the response greatly move the field forward in understanding of the systems working to produce the complicated and detrimental wasting that is cachexia. This is a work that follows all aspects in vivo in the animal model for this disease and will be very helpful in guiding future studies that work to produce therapies that can alleviate the symptoms of cachexia as an adjunct to anti tumor therapies. One of the more interesting possibilities is that by finding the PPAR link, studies on tumor models may be close to finding the common pathways that unite different tumors in the wasting phenotype. I have found the work to be entirely cohesive in design, method, reporting and conclusions and recommend immediate publication.

Reviewer #2 (Remarks to the Author):

Leukemia inhibitory factor (LIF) is a multi-functional cytokine whose overexpression is associated with poor prognosis in cancer. To address the role of LIF in cancer cachexia, the authors have generated a transgenic LIF knock-in mouse model. The authors find that LIF overexpression causes reduced energy balance in mice (as per metabolic cages) and they use metabolomics to confirm that LIF overexpression causes decreases in hepatic de novo lipogenesis and disrupted lipid homeostasis in mice. Notably, much of this signaling occurred in the liver, and the authors find that blocking LIF signaling in the liver by conditionally inducing LIFR knockout in the liver significantly attenuated LIF-induced cachexia. For the first time, the authors show that LIF signaling decreases PPAR α signaling, potentially contributing to the metabolic changes in mice. They find evidence that STAT3 phosphorylation is dramatically increased in LIF transgenic livers, and in primary mouse hepatic cells, they show that silencing STAT3 inhibits the downregulation of PPAR α by LIF. Finally, the authors show that a diet containing the PPAR α agonist fenofibrate effectively combats cachexia in mice.

Overall this is a very well done manuscript, one of the best I've seen on cachexia, and certainly the most mechanistic and rigorous. The authors use two independent systems to confirm a role for LIF in cachexia, the C26 xenograft system and their transgenic mouse. They use comprehensive metabolomics to assess

lipid changes in these systems, and they do a nice job ensuring physiologic relevance of their findings (as one example, the levels of LIF in their transgenic mice are comparable to the levels of LIF found in cachexic patients). The experimental design is comprehensive and the paper addresses an important and poorly understood topic in the cancer literature, cachexia. Addressing the following items would improve this work:

1. Unclear to this reviewer is the cell type where LIF is truly acting in the liver: this could be cleared up by some immunohistochemistry of livers from the transgenic and control mice, for pSTAT3 (and total STAT3), ACLY and PPAR α , to assess whether this signaling is occurring in hepatocytes, stellate cells, endothelial cells etc.
2. LIF is a known p53 target gene: does this mean that p53, and possibly also radiation and chemotherapy, contributes to cachexia in a p53 dependent manner? If the authors could comment on this in the Discussion, this would improve the manuscript
3. How do the authors see the regulation of STAT3 phosphorylation by LIF, can they comment on potential kinase mediators of STAT3 phosphorylation downstream of LIF? Can the authors speculate on this in the Discussion?

Reviewer #3 (Remarks to the Author):

The authors aimed to stress the importance of LIF in cancer cachexia. Although the study sheds light onto some original aspects of LIF mechanisms in the disease, some concerns arise: The abstract does not provide definition of LIF, while not characterising cachexia adequately. The introduction should provide more recent references on cachexia, as the authors refer to updated reviews(2018, 2019). The very definition of cachexia is not appropriate as fat browning is not a consensus and has never been consistently proven to occur in patients. Cachexia does not occur in up to 80% of all cancer patients, but rather 80% of all advanced stage disease patients. Maybe the authors would like to adopt the definition of cachexia as employed by the 2 international cachexia societies. There are no reports of catabolism in the brain, as stated in the introduction. L73-85 of the introduction describe what was done and found, which should be part of the discussion. The introduction should be rewritten as to describe cachexia appropriately and mention the limitations of single cytokine-targeted approaches and how the original study could add to the field. Results: 89-91, repetitive text; L107- what did the other mice die to if not cachexia or humane measures? L 159 Hypoalbuminaemia is a consistent trait of cachexia, employed even for diagnosis; how do the authors reconcile the findings? The design should include a pair-fed, not a fasted control. The results section is hard to follow as the authors mix results with detailed description of methods and sometimes, discussion. Was the fenofibrate diet consumed at the same absolute values as the control diet? Discussion, again browning is not a consensus in cachexia. L374-75: the authors cannot affirm they validated the model, since it has been long validated and employed by

other groups. Maybe they mean the model induced cachexia in their particular study, as expected, with C26 viable cells. L 376- this was reported before by others, please quote. *J Cachexia Sarcopenia Muscle* . 2018 Dec;9(6):1109-1120. doi: 10.1002/jcsm.12346. Is hepatic lipogenesis increased due to LIF or due to increased uptake from circulation, since there is lipemia as adipose tissue is wasted? STAT 3 signalling is known to change in concert with constitutive lipases and leptin signalling, yet no comment is found in discussion considering this aspect. Overall discussion does not address available literature on models and specially, in patients (there is information available for liver metabolism, circulating and hepatic TAG and lipid metabolism-related enzymes in cancer patients and animal models that were not considered).

Legends to figures are too extensive and repeat method description.

As a whole, the results may contribute to the knowledge on LIF relevance in cachexia, provided the author acquire a more comprehensive understanding of the syndrome, providing discussion of findings in accord with the more recent views, that consider tissue cross-talk. Limitations of the study should be acknowledged and stated. In summary, the design and results should be discussed bearing in mind modern concepts of cachexia (for instance, competition of tumour cells with host cells driven energy wasting is an outdated concept), specially addressing tissue cross-talk (cytokines and metabolites).

Reviewer #4 (Remarks to the Author):

Yang and colleagues investigated the effect of leukemia inhibitory factor on cachexia and hepatic lipogenesis. They claim that fenofibrate treatment could counteract LIF's adverse effects and cachexia progression. This is a potentially interesting observation. However, many problems, incomplete methodology, missing statistics, and data interpretation not supported by statistics provide an incomplete picture.

Key comments:

1. Why are serum LIF levels not detected in TgL animals? LIF is also highly expressed in adipose tissue (10.1038/s42255-021-00451-2), where it activates STAT3 signaling and modulates lipid metabolism (lipogenesis) in hepatocytes.

Indirect calorimetry, Figure 1IJK. When values are normalized per kg, it is difficult to assess the effect in cachexic mice. The same data "per mouse" should be presented in SI. No statistical details are shown for panels G-M. Where the asterisks indicate a t-test, it is not appropriate here. See DOI: 10.1038/s42255-021-00451-2.

2. Serum TG levels in absolute values (molar concentration) are critical for evaluating the cachexic profile. The heat maps could be clearer, and more information on data processing needs to be provided. Clustering statistics are missing - how did the authors decide that $C \leq 54$ and $C > 54$ is the threshold?

There is a continuous color transition. There is no biochemical rationale for the choice of these values. There is no link to lipolysis (line 206).

Furthermore, the "identification of fatty acids that form TGs", line 192, is unclear. There is no overrepresentation statistic and no method to support the selection.

3. Figure 2BC. What are the values relative to? What was the reference point? This information should be quantitative.

4. All metabolomics and lipidomics data should be annotated (Table S1 is not enough) and deposited in a repository. If the authors chase the lipidomic part, the Lipidomics Minimal Reporting Checklist should be followed (DOI 10.1038/s42255-022-00628-3). No statistics support the statement "decreased in TgLC mice", line 175.

5. de novo lipogenesis. The panels in Figure 2FG do not make sense. What are the normalized fractions? Fractions of what normalized to what? It does not fit with the methods cited. The different values for C16:0 and C18:0 are very suspicious; what about other fatty acids? What was the matrix? Is it TG pool, all lipids, or hydrolyzed fatty acids? How was the "enrichment" calculated, what was the background? This is critical information about the pathway and is not supported by data. Mice drinking 20% deuterium oxide for 7 days should be highly enriched in deuterium.

6. Figure 3E. The Western blot with n=2 is not a reliable result. If the authors want to use n=2, I expect a thorough power analysis for such a claim. The same goes for the RNA-seq data. I have never seen reasonable values with n=2. The KEGG terms are very vague (metabolic pathways), and the corresponding panels (Figure 4) show data with very variable n, why?

7. The fenofibrate experiment is based on 4-8 mice per group, while only one cohort of mice seems to be fed the diet. Why does the n vary between panels? What happened to the other mice in the group?

8. Methodology

- Mouse diet 5058 consists of preformulated hard pellets. How was fenofibrate (0.2% w/w) introduced into the diet?

- Drinking 20% D2O for 7 days is a high exposure, above the usual levels used to label liver metabolites. There is no rationale for this dose. The methodology for deuterium enrichment of lipids is completely lacking. Reference 26 used triple quadrupole MS, which could not be used to obtain valid enrichment data in this setting.

- Metabolic cages - no information on what Macro 13 means.

- Metabolomics and lipidomics - only the serum protocol is shown, while tissue/hepatocyte data were used. This needs to be included. Raw data are not stored in a public repository with metadata descriptions.

- RNA-seq method description is concise, and no data processing parameters are shown.

- Cluster 3.0 is mentioned, but no statistical results or parameters are shown to run the routines.

- Statistics are incomplete. N is very low, no power analysis was performed, Figure 1 data should be evaluated using an appropriate approach, and statistics for heat maps, clusters, etc. are missing.

Reviewer #1:

The manuscript entitled: Leukemia inhibitory factor suppresses hepatic de novo lipogenesis and induces cachexia, is a very well-constructed update to the importance of LIF to cachexia in response to tumors. The work begins by reproducing earlier studies showing the necessity of LIF to the cachexia produced by the C26 tumor. It adds the important work of showing the sufficiency of LIF to C26-induced cachexia and then takes on the underlying mechanism of the effect of LIF on adipose, the most obvious and deeply impacting tissue in the C26 cachexia disease. The findings of the importance of liver location and subsequent PPAR biology in the response greatly move the field forward in understanding of the systems working to produce the complicated and detrimental wasting that is cachexia. This is a work that follows all aspects in vivo in the animal model for this disease and will be very helpful in guiding future studies that work to produce therapies that can alleviate the symptoms of cachexia as an adjunct to anti-tumor therapies. One of the more interesting possibilities is that by finding the PPAR link, studies on tumor models may be close to finding the common pathways that unite different tumors in the wasting phenotype. I have found the work to be entirely cohesive in design, method, reporting and conclusions and recommend immediate publication.

Response: We want to thank the reviewer for such positive and encouraging comments about our work. We appreciate that the reviewer pointed out the potential development of this finding for future application in cancer patients, as well as the future direction to follow up on the PPAR α link in cachexia, which we will investigate in our future study. Thank you!

Reviewer #2:

Leukemia inhibitory factor (LIF) is a multi-functional cytokine whose overexpression is associated with poor prognosis in cancer. To address the role of LIF in cancer cachexia, the authors have generated a transgenic LIF knock-in mouse model. The authors find that LIF overexpression causes reduced energy balance in mice (as per metabolic cages) and they use metabolomics to confirm that LIF overexpression causes decreases in hepatic de novo lipogenesis and disrupted lipid homeostasis in mice. Notably, much of this signaling occurred in the liver, and the authors find that blocking LIF signaling in the liver by conditionally inducing LIFR knockout in the liver significantly attenuated LIF-induced cachexia. For the first time, the authors show that LIF signaling decreases PPAR α signaling, potentially contributing to the metabolic changes in mice. They find evidence that STAT3 phosphorylation is dramatically increased in LIF transgenic livers, and in primary mouse hepatic cells, they show that silencing STAT3 inhibits the downregulation

of PPAR α by LIF. Finally, the authors show that a diet containing the PPAR α agonist fenofibrate effectively combats cachexia in mice.

Overall this is a very well done manuscript, one of the best I've seen on cachexia, and certainly the most mechanistic and rigorous. The authors use two independent systems to confirm a role for LIF in cachexia, the C26 xenograft system and their transgenic mouse. They use comprehensive metabolomics to assess lipid changes in these systems, and they do a nice job ensuring physiologic relevance of their findings (as one example, the levels of LIF in their transgenic mice are comparable to the levels of LIF found in cachexic patients). The experimental design is comprehensive and the paper addresses an important and poorly understood topic in the cancer literature, cachexia. Addressing the following items would improve this work:

1. Unclear to this reviewer is the cell type where LIF is truly acting in the liver: this could be cleared up by some immunohistochemistry of livers from the transgenic and control mice, for pSTAT3 (and total STAT3), ACLY and PPAR α , to assess whether this signaling is occurring in hepatocytes, stellate cells, endothelial cells etc.

Response: Thank the reviewer for this very good suggestion. While hepatocytes are the predominant cell type in the liver, additional cell types exist in the liver. As suggested by the reviewer, we performed immunohistochemistry (IHC) assays using pSTAT3, total STAT3, PPAR α , and ACLY antibodies in the liver tissues from mice with or without LIF overexpression (e.g. TgLC and TgL mice with TAM injection). As shown in **Fig. S5B & S6A**, we observed increased nuclear pSTAT3 levels, decreased PPAR α and ACLY levels predominantly in hepatocytes in the liver from TgLC mice with TAM injection compared with TgL mice with TAM injection. No obvious differences in total STAT3 levels were observed between TgLC and TgL mice with TAM injection. These observations were consistent with the results obtained from the Western blot assays (**Fig. 4H & 5A**) and further indicated that LIF acts predominantly on hepatocytes in the liver.

2. LIF is a known p53 target gene: does this mean that p53, and possibly also radiation and chemotherapy, contributes to cachexia in a p53 dependent manner? If the authors could comment on this in the Discussion, this would improve the manuscript.

Response: Thank the reviewer for this very good question. Our previous study identified LIF as a p53 target gene (Hu et al., 2007). p53 regulates the expression of basal as well as inducible levels of LIF under certain stress conditions, including radiation and chemotherapy, in several cell types. In addition, the expression of LIF is regulated by other factors under different conditions, and the regulation of LIF is highly context-dependent (Wang et al., 2023; Zhang et al., 2021). As a

pleiotropic cytokine, LIF exhibits different functions in a highly context-dependent manner. The cachexia induced by LIF, as described in this study, requires the continuous systemic elevation of LIF at high levels. It is currently unclear whether and how the regulation of LIF by p53, as well as other regulators of LIF, play a role in cachexia, which warrants further investigation in future studies. We have added these discussions in the revised manuscript (Page 22, lines 455-460).

3. How do the authors see the regulation of STAT3 phosphorylation by LIF, can they comment on potential kinase mediators of STAT3 phosphorylation downstream of LIF? Can the authors speculate on this in the Discussion?

Response: Thank the reviewer for this very good question. The binding of LIF to the LIFR induces the heterodimerization of LIFR with gp130. The formation of this receptor complex results in the activation of the receptor-associated Janus kinases (JAKs) by phosphorylating receptor docking sites, which in turn leads to the recruitment and activation of Src Homology-2 (SH2) domain-containing proteins, such as STAT3 (Viswanadhapalli et al., 2022; Wang et al., 2023). We have added this to the Discussion section (Page 19, lines 390-393).

Reviewer #3:

The authors aimed to stress the importance of LIF in cancer cachexia. Although the study sheds light onto one original aspects of LIF mechanisms in the disease, some concerns arise:

1. The abstract does not provide definition of LIF, while not characterising cachexia adequately. The introduction should provide more recent references on cachexia, as the authors refer to updated reviews (2018, 2019). The very definition of cachexia is not appropriate as fat browning is not a consensus and has never been consistently proven to occur in patients. Cachexia does not occur in up to 80% of all cancer patients, but rather 80% of all advanced stage disease patients. Maybe the authors would like to adopt the definition of cachexia as employed by the 2 international cachexia societies. There are no reports of catabolism in the brain, as stated in the introduction. L73-85 of the introduction describe what was done and found, which should be part of the discussion. The introduction should be rewritten as to describe cachexia appropriately and mention the limitations of single cytokine-targeted approaches and how the original study could add to the field.

Response: Thank the reviewer for these very good suggestions. As suggested by the reviewer, we provided the definition of LIF in the Abstract. In both the Abstract and Introduction sections, we have revised the definition of cachexia. We have added two recent excellent references (Ferrer M., et al., Cell 2023, 186:1824; Pryce B., et al., Cancer Cell 2023, 41:581) reflecting the definition of cachexia employed by the 2 international cachexia societies (the Cancer Cachexia Society and the Cancer Cachexia Action Network). As suggested by the reviewer, we have removed the statements about “the fat browning” and “catabolism in the brain”. We further included the limitations of single cytokine-targeted approaches and described how this study could add to the field in the Introduction section.

2. Results: 89-91, repetitive text. The results session is hard to follow as the authors mixes results with detailed description of methods and sometimes, discussion.

Response: As suggested by the reviewer, we have removed the repetitive text. We have deleted some detailed descriptions of methods in the Results section.

3. L107- what did the other mice die to if not cachexia or humane measures?

Response: The majority of mice bearing C26-LIF KO tumors reached the humane endpoint due to the size of the tumors, while a small percentage of mice were due to cachexia. We revised this sentence in Page 6, lines 120-121 to increase the clarity.

4. L 159 Hypoalbuminaemia is a consistent trait of cachexia, employed even for diagnosis; how do the authors reconcile the findings?

Response: Thank the reviewer for this very insightful question. Studies have shown that patients with cachexia have lower albumin levels than those without cachexia (Guo et al., 2022; Lai et al., 2022; Liu et al., 2021). Hypoalbuminemia can indicate the onset of cachexia and is an early prognostic marker for poor prognosis (Guo et al., 2022; Lai et al., 2022; Liu et al., 2021). By measuring serum albumin levels in TgLC mice at 3 days after TAM injection, when mice started to exhibit cachexia phenotypes, we observed a significant decrease in serum albumin levels. The results in **Fig. 1N** in the original submission, which showed increased serum albumin levels in TgLC mice, were obtained from mice at 9 days after TAM injection. TgLC mice exhibited very severe cachexia phenotypes at 9 days after TAM injection, and the median survival of TgLC mice with TAM injection was 10 days. The increased serum albumin levels in these mice might reflect severe dehydration in these mice. We have updated the results of serum albumin levels in **Fig. 1N** to more accurately reflect its change and its correlation with cachexia development.

5. The design should include a pair-fed, not a fasted control.

Response: Thank the reviewer for this very good suggestion. Pair-feeding by restricting the food intake of TgL mice with TAM injection to match that of TgLC with TAM injection can eliminate differences caused by the total amount of food consumed. As suggested by the reviewer, we performed the following experiments including pair-feeding in the design.

A) We examined the change in mouse body weight and body composition in the pair-fed TgL mice with TAM injection. As shown in **Fig. S2A & B**, the pair-fed TgL mice with TAM injection exhibited a slight trend of body weight and lean mass decrease, along with a significant decrease of fat mass compared with the regular-fed TgL mice with TMA injection. However, the decrease in body weight, lean mass and fat mass in the pair-fed TgL mice with TAM injection was significantly less than the decrease of these parameters in TgLC mice with TAM injection.

B) We observed that LIF overexpression activates the STAT3 signaling to downregulate the expression of PPAR α in TgLC mice with TAM injection. As a control, we examined the effect of pair-feeding on the activity of the STAT3 signaling and the expression of PPAR α by examining the levels of pSTAT3 at Tyr705 using Western blot assays and the mRNA levels of PPAR α using real-time PCR assays, respectively. No obvious activation of the STAT3 signaling and no significant change in the mRNA levels of PPAR α were observed in the liver from the pair-fed TgL mice with TAM injection (**Fig. S5C, Fig. S6B**).

The results from this set of pair-feeding experiments demonstrate the contribution of reduced food intake to the change of body weight, and at the same time show that the changes in the STAT3 signaling activity and PPAR α expression levels in the liver of TgLC mice with LIF overexpression are not solely attributed to the reduced food intake of these mice.

6. Was the fenofibrate diet consumed at the same absolute values as the control diet?

Response: Thank the reviewer for this very important question. As suggested by the reviewer, we measured the food intake in TgLC mice with TAM injection fed with regular chow or fenofibrate diet. As shown in **Fig. S7C**, TgLC mice with TAM injection fed with fenofibrate diet showed a trend of improved food intake compared with mice fed with regular chow.

7. Discussion, again browning is not a consensus in cachexia.

Response: Thank the reviewer for this important point. We have removed “fat browning” from the definition of cachexia in the Discussion section.

8. L374-75: *the authors cannot affirm they validated the model, since it has been long validated and employed by other groups. Maybe they mean the model induced cachexia in their particular study, as expected, with C26 viable cells. L 376- this was reported before by others, please quote. J Cachexia Sarcopenia Muscle. 2018 Dec;9(6):1109-1120. doi: 10.1002/jcsm.12346.*

Response: We agree with the reviewer that it is more appropriate to state that “This study observes that C26 tumor-bearing mice exhibit body weight loss and poor survival, which is consistent with previous reports.” We have revised this statement in the Discussion section (Page 19, lines 402-404). We have also quoted the important reference on C26-induced cachexia, as the reviewer suggested.

9. *Is hepatic lipogenesis increased due to LIF or due to increased uptake from circulation, since there is lipemia as adipose tissue is wasted?*

Response: Thank the reviewer for raising this important question. One important finding of this study is the change in hepatic lipogenesis. The results in this study (**Fig. 2F & G**) showed that LIF overexpression decreases hepatic *de novo* lipogenesis in mice. Furthermore, the results in this study showed that LIF overexpression activates the STAT3 signaling to downregulate PPAR α , which in turn decreases the expression of a list of PPAR α target genes involved in lipogenesis (**Fig. 4G, 4H & 5**). This represents an important mechanism for the decreased hepatic lipogenesis induced by LIF overexpression.

10. *STAT 3 signalling is known to change in concert with constitutive lipases and leptin signalling, yet no comment is found in discussion considering this aspect. Overall discussion does not address available literature on models and specially, in patients (there is information available for liver metabolism, circulating and hepatic TAG and lipid metabolism-related enzymes in cancer patients and animal models that were not considered).*

Response: Thank the reviewer for this important suggestion. As suggested by the reviewer, we have discussed the connection among the STAT3 signaling, lipases and the leptin levels (Page 21, lines 443-447). We have also discussed the metabolic changes, including the changes in circulating and hepatic TG levels, reported in the literature (Page 20, lines 422-424; Page 21 lines 431-433).

11. *Legends to figures are too extensive and repeat method description.*

Response: Figure legends have been revised to remove too extensive description about methods.

As a whole, the results may contribute to the knowledge on LIF relevance in cachexia, provided the author acquire a more comprehensive understanding of the syndrome, providing discussion of findings in accord with the more recent views, that consider tissue cross-talk. Limitations of the study should be acknowledged and stated. In summary, the design and results should be discussed bearing in mind modern concepts of cachexia (for instance, competition of tumour cells with host cells driven energy wasting is an outdated concept), specially addressing tissue cross-talk (cytokines and metabolites).

Response: We thank the reviewer for the positive comments on the contribution of this study to the knowledge on LIF in cachexia. We also thank the reviewer for all these constructive and insightful suggestions to help improving the manuscript. By addressing the above specific comments and revising the manuscript accordingly, we believe that the revised manuscript presents a more carefully characterized cachexia syndrome, and is able to better reflect the more recent views on cachexia with a more through discussion.

Reviewer #4:

Yang and colleagues investigated the effect of leukemia inhibitory factor on cachexia and hepatic lipogenesis. They claim that fenofibrate treatment could counteract LIF's adverse effects and cachexia progression. This is a potentially interesting observation. However, many problems, incomplete methodology, missing statistics, and data interpretation not supported by statistics provide an incomplete picture.

Key comments:

1. Why are serum LIF levels not detected in TgL animals? LIF is also highly expressed in adipose tissue (10.1038/s42255-021-00451-2), where it activates STAT3 signaling and modulates lipid metabolism (lipogenesis) in hepatocytes. Indirect calorimetry, Figure 11JK. When values are normalized per kg, it is difficult to assess the effect in cachexic mice. The same data "per mouse" should be presented in SI. No statistical details are shown for panels G-M. Where the asterisks indicate a t-test, it is not appropriate here. See DOI: 10.1038/s42255-021-00451-2.

Response: Thank the reviewer for this set of important questions. TgL mice contain a knock-in of the mouse LIF gene, preceded by the CAG promoter and a transcriptional STOP cassette in the Rosa26 locus. TgL mice don't possess the Cre allele to induce LIF overexpression. Serum LIF levels in TgL mice reflect basal serum LIF levels in mice under physiological conditions, which are below the detection limitation of the ELISA kit used in this study (mouse LIF DuoSet kit, R&D system).

While the basal LIF levels in the serum and adipose tissues are generally very low, increases in LIF levels in adipose tissue and serum have been observed in pre-clinical mouse models and patients with obesity, as well as in cachexia (Arora et al., 2018; Onate et al., 2013; Yuan et al., 2022). A very recent study showed that the obesity-associated LIF receptor (LIFR)/STAT3 signaling in adipocytes can modulate lipid metabolism in the liver and contribute to liver triglyceride (TG) accumulation (Guo et al., 2021), which highlights the importance of the cytokine induced inter-tissue crosstalk in metabolic dysregulation and cachexia development. The TgLC and LIFR^{fllox/fllox} models can be used to study and understand the crosstalk among different tissues during cachexia development in future studies. We have added these discussions in the Discussion section (Page 22, lines 463-467).

As suggested by the reviewer, we have presented the same data for **Fig. 11JK** calculated as "per mice" in **Fig. S2D, F & H**, which show the similar trend to that observed in **Fig. 11JK**.

For **Fig. 1G-M**, two-tailed Student's *t*-test was used to compare the statistical differences between TgL and TgLC mice at different time points after TAM injection. As suggested by the reviewer, we performed statistical analysis on the data in **Fig. 1G-M** using the linear regression model, a procedure equivalent to ANCOVA. As the measurements on food intake, TEE, VO₂, and VCO₂ are longitudinal data, we used the generalized estimating equation (GEE) method with

Gaussian distribution to analyze the data to account for the within-animal correlation. We specified a linear regression model for each outcome with the group and day interaction terms while adjusting for animal weight. Estimated by the GEE method, the within-animal correlation structure was specified using the first-order autoregressive correlation. We performed the analysis using the geepack package in R. The results of GEE analysis are presented in **Supplementary Material**. The analysis for food intake, TEE, VO₂ and VCO₂ of TgL and TgLC mice post TAM injection was shown in **Fig. S2C, E, G & I**, and very similar statistical results were obtained compared with Student's *t*-test used in **Fig. 1G-M**.

2. Serum TG levels in absolute values (molar concentration) are critical for evaluating the cachexic profile. The heat maps could be clearer, and more information on data processing needs to be provided. Clustering statistics are missing - how did the authors decide that $C \leq 54$ and $C > 54$ is the threshold? There is a continuous color transition. There is no biochemical rationale for the choice of these values. There is no link to lipolysis (line 206). Furthermore, the "identification of fatty acids that form TGs", line 192, is unclear. There is no overrepresentation statistic and no method to support the selection.

Response: Thank the reviewer for this very important question. We completely agree with the reviewer that it is critical to use absolute values of TG levels to evaluate the cachexic profile. The Heatmaps of TG levels in this study, including **Fig. 2A, D, E & 6E**, were obtained by using the absolute values from lipidomics analysis. As suggested by the reviewer, we have provided the information on data processing. The Heatmap was obtained by performing Hierarchical Clustering analysis of raw data of TGs using Clster 3.0 and visualized *via* Treeview. Results showed that

TGs were clustered into 2 groups, separated by the length of the carbon chain; one group contains TGs with a carbon chain equal to or less than 54 ($C \leq 54$), and the other group contains TGs with a carbon chain larger than 54 ($C > 54$). To increase clarity, we have presented the Treeview for Heatmaps in **Fig. 2A, D & E**.

We agree with the reviewer that there is no link to lipolysis in line 206 of the original version of the manuscript, and we have removed this statement.

In line 192, the annotation of TGs was performed using MS-DIAL. We have completed the Lipidomics Reporting Checklist to disclose the full details of the method.

3. *Figure 2BC. What are the values relative to? What was the reference point? This information should be quantitative.*

Response: Thank the reviewer for this question. For **Fig. 2B**, the TG levels under the fed condition are designated as 1. For **Fig. 2C**, the TG levels in TgL mice under the fed condition are designated as 1. We have added this description to the figure legend to increase clarity. The raw data has been uploaded to Metabolomics Workbench with the following access links:

<http://dev.metabolomicsworkbench.org:22222/data/DRCCMetadata.php?Mode=Study&StudyID=ST002842&Access=OvpY4776>;

<http://dev.metabolomicsworkbench.org:22222/data/DRCCMetadata.php?Mode=Study&StudyID=ST002843&Access=IimV6908>;

<http://dev.metabolomicsworkbench.org:22222/data/DRCCMetadata.php?Mode=Study&StudyID=ST002844&Access=RycF4420>);

<http://dev.metabolomicsworkbench.org:22222/data/DRCCMetadata.php?Mode=Study&StudyID=ST002824&Access=ZioZ3524>

4. *All metabolomics and lipidomics data should be annotated (Table S1 is not enough) and deposited in a repository. If the authors chase the lipidomic part, the Lipidomics Minimal Reporting Checklist should be followed (DOI 10.1038/s42255-022-00628-3). No statistics support the statement "decreased in TgLC mice", line 175.*

Response: Thank the reviewer for this important suggestion. As suggested by the reviewer, we have uploaded the annotated data and the raw data of metabolomics to Metabolomics Workbench. We have also completed the Lipidomics Minimal Reporting Checklist on the Lipidomics Standard Initiative (LSI) website using the link provided by the editor. The checklist is included as **Supplementary Material**.

For line 175 in the original submission, we have revised the statement about the difference in several metabolites observed in **Fig. S3B** as “a trend of decrease”.

5. *de novo* lipogenesis. The panels in Figure 2FG do not make sense. What are the normalized fractions? Fractions of what normalized to what? It does not fit with the methods cited. The different values for C16:0 and C18:0 are very suspicious; what about other fatty acids? What was the matrix? Is it TG pool, all lipids, or hydrolyzed fatty acids? How was the “enrichment” calculated, what was the background? This is critical information about the pathway and is not supported by data. Mice drinking 20% deuterium oxide for 7 days should be highly enriched in deuterium.

Response: Thank the reviewer for this set of important questions. To increase clarity and avoid confusion, we have changed the labeling of the Y-axis of **Fig. 2F & G** to “²H-labeled fraction (%)”. The data in **Fig. 2F & G** showed the percentage of ²H-labeled C16:0 and C18:0 of the total C16:0 and C18:0, respectively, which was calculated after the ¹³C natural abundance correction. To assess *de novo* lipogenesis, we extracted and saponified the lipids to release the fatty acyl groups. The detected fatty acids were produced after saponification and dissolved in a 1:1 isopropanol:methanol solution. Detailed methods have been provided in the Methods section (Page 25, lines 507-520).

C16:0 and C18:0 were chosen for analysis in this study for the following two reasons. Firstly, C16:0 and C18:0 are much more abundant than other fatty acids, allowing for more accurate detection of ²H incorporation compared with other fatty acids. Secondly, ²H incorporation in saturated fatty acids, such as C16:0 and C18:0, can better demonstrate *de novo* lipogenesis activity, whereas the production of unsaturated fatty acids is also affected by the activity of desaturases that convert saturated fatty acids into unsaturated fatty acids and polyunsaturated fatty acids (PUFAs).

While we didn’t measure the enrichment of D₂O in body water in this study, it has been reported previously that mice drinking 25-30% D₂O for 8-21 days had an enrichment of D₂O in body water at a level of 15-17.5% (Jandova et al., 2023; Shi et al., 2018). In mice, D₂O enrichment below 20% does not cause any obvious effect on physiological processes and is considered a safe level (Jones and Leatherdale, 1991; Katz et al., 1962). Yet, a low D₂O concentration (~15%) means that deuterium could only sparsely label newly synthesized macromolecules (Shi et al., 2018). Therefore, we chose the condition of giving mice 20% D₂O in drinking water for 7 days to detect *de novo* lipogenesis.

6. Figure 3E. The Western blot with n=2 is not a reliable result. If the authors want to use n=2, I expect a thorough power analysis for such a claim. The same goes for the RNA-seq data. I have never seen reasonable values with n=2. The KEGG terms are very vague (metabolic pathways), and the corresponding panels (Figure 4) show data with very variable n, why?

Response: As suggested by the reviewer, we examined pSTAT3 and total STAT3 protein levels in additional samples (n=3/group) using Western blot assays and obtained consistent results (**Fig. S4**). We also obtained RNA-seq data from additional samples to increase the sample size for each group (n=3 for TgL mouse with TAM injection and n=4 for TgLC mice with TAM injection) and updated the results in **Fig. 4**.

KEGG analysis of DEGs was performed using the Database for Annotation, Visualization and Integrated Discovery (DAVID), with a higher enrichment score signifying more cluster enrichment. Terms of the clusters, including metabolic pathways, were provided by DAVID.

Data in different panels in **Fig. 4** were obtained from different individual biological repeats from multiple batches of experiments, resulting in varying n values among different panels.

7. *The fenofibrate experiment is based on 4-8 mice per group, while only one cohort of mice seems to be fed the diet. Why does the n vary between panels? What happened to the other mice in the group?*

Response: In the fenofibrate experiment (**Fig. 6**), data from different panels examining different parameters were often obtained from different batches of experiments using different cohorts of mice due to the differences in experimental settings. Therefore, n varies among different panels.

8. Methodology

- *Mouse diet 5058 consists of preformulated hard pellets. How was fenofibrate (0.2% w/w) introduced into the diet?*

Response: Thank the reviewer for this important question. Fenofibrate (0.2% w/w) diet was custom-made by TestDiet, a company specializing in aiding researchers who need modifications to standard LabDiet formulas. This information has been added to the Method section (Page 24, lines 499-500).

- *Drinking 20% D2O for 7 days is a high exposure, above the usual levels used to label liver metabolites. There is no rationale for this dose. The methodology for deuterium enrichment of lipids is completely lacking. Reference 26 used triple quadrupole MS, which could not be used to obtain valid enrichment data in this setting.*

Response: Thank the reviewer for this set of important questions. As explained in our response to comment #5, previous studies have shown that mice drinking 25-30% D₂O for 8-21 days had an enrichment of D₂O in body water at a level of 15-17.5% (Jandova et al., 2023; Shi et al., 2018), which is considered safe and suitable for sparsely labeling newly synthesized macromolecules (Shi et al., 2018). Therefore, the condition we used in this study (20% D₂O in drinking water for 7 days) is appropriate for detecting *de novo* lipogenesis.

We have also provided the description of measuring deuterium enrichment of lipids in the Method section (Page 25, lines 507-520).

As suggested by the reviewer, we have removed Reference 26.

- *Metabolic cages - no information on what Macro 13 means.*

Response: Thank the reviewer for raising the question about Macro 13. Macro 13 is a built-in macro function in the Promethion software package designed to process the data collected by Promethion systems. Its primary function is to produce standardized output formats for the metabolic variables of interest at each cage. The processed data generated by Macro 13 were then analyzed using the CalR software. We have added these details about Macro 13 in the Methods section (Page 26, lines 525-529).

- *Metabolomics and lipidomics - only the serum protocol is shown, while tissue/hepatocyte data were used. This needs to be included. Raw data are not stored in a public repository with metadata descriptions.*

Response: Thank the reviewer for this question. The protocol for metabolite extraction from serum and tissues is detailed in the Methods section (Page 27, lines 547-561). Line 547 describes the procedure for serum and line 548 describes the procedure for tissues. Lines 549-561 describes the following steps that are the same for both serum and tissue samples.

The raw data of metabolomics have been uploaded to Metabolomics Workbench, with the access links provided in the response to comment #3 of Reviewer #4.

- RNA-seq method description is concise, and no data processing parameters are shown.

Response: As suggested by the reviewer, we have added a detailed description of RNA-seq in the Methods section (Page 29, lines 602-607) as following. “Raw data (raw reads) in fastq format were initially processed using the fastp software, and then mapped to the mouse reference genome. Differential expression analysis was performed using the DESeq2 method (Love et al., 2014). False discovery rate (FDR) was used to control for the multiple comparisons based on the Benjamini and Hochberg method. An FDR cutoff of 0.05 and an absolute fold change of 2 were set as the threshold to select for genes with significant differential expression.”

- Cluster 3.0 is mentioned, but no statistical results or parameters are shown to run the routines.

Response: As suggested by the reviewer, we have provided details about the clustering analysis using Cluster 3.0 in the Methods section (Pages 29-30, lines 612-615) as following. “Heatmap was generated using the Cluster 3.0 software and visualized *via* Treeview as described previously (de Hoon et al., 2004; Eisen et al., 1998). Briefly, raw data were first converted to Log transform data and then used for Hierarchical Clustering analysis to generate the cdt file, which was subsequently visualized with Treeview.”

- Statistics are incomplete. N is very low, no power analysis was performed, Figure 1 data should be evaluated using an appropriate approach, and statistics for heat maps, clusters, etc. are missing.

Response: Thank the reviewer for this important question. As our responses to comments #1, 2, 6 and some comments in #8, we have increased “n” in several experiments, performed appropriate statistical analysis as suggested by the reviewer, provided detailed information for Heatmaps, clusters, etc. These added data, information and statistical analysis provide appropriate statistical analysis for this study.

Again, we want to thank the reviewers for your great efforts and very insightful and constructive comments to improve our manuscript. We hope that with these changes and added experiments, our manuscript would be acceptable for publication. Thank you very much!

References

- Arora, G.K., Gupta, A., Narayanan, S., Guo, T., Iyengar, P., and Infante, R.E. (2018). Cachexia-associated adipose loss induced by tumor-secreted leukemia inhibitory factor is counterbalanced by decreased leptin. *JCI Insight* 3.
- de Hoon, M.J., Imoto, S., Nolan, J., and Miyano, S. (2004). Open source clustering software. *Bioinformatics* 20, 1453-1454.
- Eisen, M.B., Spellman, P.T., Brown, P.O., and Botstein, D. (1998). Cluster analysis and display of genome-wide expression patterns. *Proc Natl Acad Sci U S A* 95, 14863-14868.
- Guo, T., Gupta, A., Yu, J., Granados, J.Z., Gandhi, A.Y., Evers, B.M., Iyengar, P., and Infante, R.E. (2021). LIFR- α -dependent adipocyte signaling in obesity limits adipose expansion contributing to fatty liver disease. *iScience* 24, 102227.
- Guo, Y., Wei, L., Patel, S.H., Lopez, G., Grogan, M., Li, M., Haddad, T., Johns, A., Ganesan, L.P., Yang, Y., *et al.* (2022). Serum Albumin: Early Prognostic Marker of Benefit for Immune Checkpoint Inhibitor Monotherapy But Not Chemoimmunotherapy. *Clin Lung Cancer* 23, 345-355.
- Hu, W., Feng, Z., Teresky, A.K., and Levine, A.J. (2007). p53 regulates maternal reproduction through LIF. *Nature* 450, 721-724.
- Jandova, J., Galons, J.P., Dettman, D.L., and Wondrak, G.T. (2023). Systemic deuteration of SCID mice using the water-isotopologue deuterium oxide (D(2) O) inhibits tumor growth in an orthotopic bioluminescent model of human pancreatic ductal adenocarcinoma. *Mol Carcinog* 62, 598-612.
- Jones, P.J., and Leatherdale, S.T. (1991). Stable isotopes in clinical research: safety reaffirmed. *Clin Sci (Lond)* 80, 277-280.
- Katz, J.J., Crespi, H.L., Czajka, D.M., and Finkel, A.J. (1962). Course of deuteration and some physiological effects of deuterium in mice. *Am J Physiol* 203, 907-913.
- Lai, K.C., Hong, Z.X., Hsieh, J.G., Lee, H.J., Yang, M.H., Hsieh, C.H., Yang, C.H., and Chen, Y.R. (2022). IFIT2-depleted metastatic oral squamous cell carcinoma cells induce muscle atrophy and cancer cachexia in mice. *J Cachexia Sarcopenia Muscle* 13, 1314-1328.
- Liu, X.Y., Zhang, X., Ruan, G.T., Zhang, K.P., Tang, M., Zhang, Q., Song, M.M., Zhang, X.W., Ge, Y.Z., Yang, M., *et al.* (2021). One-Year Mortality in Patients with Cancer Cachexia: Association with Albumin and Total Protein. *Cancer Manag Res* 13, 6775-6783.
- Love, M.I., Huber, W., and Anders, S. (2014). Moderated estimation of fold change and dispersion for RNA-seq data with DESeq2. *Genome Biol* 15, 550.
- Onate, B., Vilahur, G., Camino-Lopez, S., Diez-Caballero, A., Ballesta-Lopez, C., Ybarra, J., Moscatiello, F., Herrero, J., and Badimon, L. (2013). Stem cells isolated from adipose tissue of obese patients show changes in their transcriptomic profile that indicate loss in stemcellness and increased commitment to an adipocyte-like phenotype. *BMC Genomics* 14, 625.
- Shi, L., Zheng, C., Shen, Y., Chen, Z., Silveira, E.S., Zhang, L., Wei, M., Liu, C., de Sena-Tomas, C., Targoff, K., *et al.* (2018). Optical imaging of metabolic dynamics in animals. *Nat Commun* 9, 2995.

- Viswanadhapalli, S., Dileep, K.V., Zhang, K.Y.J., Nair, H.B., and Vadlamudi, R.K. (2022). Targeting LIF/LIFR signaling in cancer. *Genes Dis* 9, 973-980.
- Wang, J., Chang, C.Y., Yang, X., Zhou, F., Liu, J., Feng, Z., and Hu, W. (2023). Leukemia inhibitory factor, a double-edged sword with therapeutic implications in human diseases. *Mol Ther* 31, 331-343.
- Yuan, Y., Li, K., Teng, F., Wang, W., Zhou, B., Zhou, X., Lin, J., Ye, X., Deng, Y., Liu, W., *et al.* (2022). Leukemia inhibitory factor protects against liver steatosis in nonalcoholic fatty liver disease patients and obese mice. *J Biol Chem* 298, 101946.
- Zhang, C., Liu, J., Wang, J., Hu, W., and Feng, Z. (2021). The emerging role of leukemia inhibitory factor in cancer and therapy. *Pharmacol Ther* 221, 107754.

REVIEWERS' COMMENTS

Reviewer #2 (Remarks to the Author):

The authors have addressed all of my previous concerns, the result is an extremely well done and interesting manuscript

Reviewer #3 (Remarks to the Author):

The authors have addressed the concerns mentioned in the review process, and included new data. The revised manuscript includes paired animal results, and the main points causing confusion were clarified as requested. I recommend the publication of the manuscript after revision of grammar.

Line 321 "obvious changes". I believe authors mean conspicuous or major changes, instead.

Reviewer #4 (Remarks to the Author):

The authors significantly updated the manuscript and responded to my comments. However, I consider the story to be of an evolutionary nature.

Reviewer #2:

The authors have addressed all of my previous concerns, the result is an extremely well done and interesting manuscript

Reviewer #3:

The authors have addressed the concerns mentioned in the review process, and included new data. The revised manuscript includes paired animal results, and the main points causing confusion were clarified as requested. I recommend the publication of the manuscript after revision of grammar. Line 321 "obvious changes". I believe authors mean conspicuous or major changes, instead.

Reviewer #4:

The authors significantly updated the manuscript and responded to my comments. However, I consider the story to be of an evolutionary nature.

Response: Thank all three reviewers for very positive comments. In response to Reviewer #3's comments, we requested Jennifer Hostettle of the Medical Writing Services of Rutgers Cancer Institute of New Jersey to help for revision of grammar, and her assistance has been acknowledged in the Acknowledgment session.

Again, we want to thank reviewers for their great efforts and very insightful and constructive comments to improve our manuscript. Thank you very much!